# Dynamic Modeling and Attitude–Vibration Cooperative Control for a Large-Scale Flexible Spacecraft

Guiqin He and Dengqing Cao *

School of Astronautics, Harbin Institute of Technology, Harbin 150001, China
* Correspondence: dqcao@hit.edu.cn

**Abstract:** Modern spacecraft usually have larger and more flexible appendages whose vibration becomes more and more prominent, and it has a great influence on the precision of spacecraft attitude. Therefore, the cooperative control of attitude maneuvering and structural vibration of the system has become a significant issue in the spacecraft design process. We developed a low-dimensional and high-precision mathematical model for a large-scale flexible spacecraft (LSFS) equipped with a pair of hinged solar arrays in this paper. The analytic global modes are used to obtain the rigid–flexible coupling discrete dynamic model, and the governing equations with multiple DOFs for the system are derived by using the Hamiltonian principle. The rigid–flexible coupled oscillating responses of LSFS under the three-axis attitude-driving torque pulse during the in-orbit attitude maneuvering process are investigated. A study on the flexibility of the hinge was also conducted. Based on the simplified and accurate dynamic model of the system, we can obtain a state-space model for LSFS conveniently, and the cooperative control schemes for rigid motion and flexible oscillation control are designed by using the LQR, PD, and PD + IS algorithms. The simulation results show that three cooperative controllers can realize spacecraft attitude adjustment and synchronously eliminate flexible oscillation successfully. By comparison, the PD + IS controller is simpler so that it is suitable for the real-time attitude–vibration cooperative control of spacecraft.

**Keywords:** large-scale flexible spacecraft; hinged multi-panel solar array; low-dimensional and high-precision model; attitude maneuvering; cooperative control

## 1. Introduction

A large-scale flexible spacecraft (LSFS) employed for communications, remote sensing, or other applications typically has large-span jointed solar arrays, which provide enough power for these spacecraft to perform their various functions and ensure that the spacecraft's lifetime is sufficiently long. Due to the large scale of solar arrays, the rigid–flexible coupling vibration of these extremely flexible spacecraft can be easily triggered by operations in orbit such as the attitude maneuver [1]. Spacecraft in orbit are also affected by their own complex environmental factors, especially large flexible spacecraft in the course of performing diversified missions. This process can easily trigger the vibration of large flexible appendages. However, because spacecraft are not affected by air resistance in space, the low-frequency vibration of flexible structures is not likely to attenuate, and the vibration of flexible appendages can have a coupling effect with spacecraft attitude. Moreover, it can cause irreversible damage to precise parts of spacecraft and even cause the spacecraft to roll unsteadily, leading to failure. Therefore, in order to investigate the dynamic characteristics and design an effective attitude–vibration cooperative control law for LSFS, obtaining their analytic global rigid–flexible coupling modes and establishing dynamic models with low degrees of freedom is of great importance.

As vital appendages of spacecraft, solar arrays typically consist of flexible solar panels and hinges. These flexible hinges have a huge influence on the whole spacecraft's dynamics. The dynamics of spacecraft hinged structures, especially the flexibility and non-linearity

of the hinge, have an important influence on the dynamics of the system. Therefore, the study of spacecraft hinged structure dynamics is an important part of spacecraft design. It is directly related to the determination of spacecraft configuration, control, and so on, and is one of the key issues to be considered when designing spacecraft. During spacecraft maneuvering, structural vibrations are unavoidable due to the extreme flexibility of the structure. Hence, the simplification of flexure hinges is critical to the dynamics and control of LSFS. For such multi-body structures connected with flexible hinges, there are several common methods. (1) The hinge is equivalent to a single beam structure. (2) The hinge is considered as a rotational spring. (3) The connection structure is simplified into a spring-damping element. (4) The hinge is characterized as a spring-constrained pendulum structure, and the mechanical model of the hinge is obtained by introducing contact theory. Wei and Cao [2] proposed the global mode method to establish an analytical dynamic model for the flexible spacecraft with hinged appendages. Further, Wei et al. [3] introduced nonlinear joints in a multi-beam structure and studied the complicated nonlinear responses of the system. They focused on the strong impact of flexible hinges on the dynamic behavior of the whole system. The good results demonstrate that the global mode method is efficient to solve dynamic modeling problems of hinged structures. In fact, as we know, the exact solution for the plate dynamic problem is always difficult to find except in some typical and classical cases with simple boundaries. In the case of solar arrays, how to find the exact solutions for complex boundaries connected by hinges is still a tough issue in the mechanic field. Xing and Liu [4] developed a novel method to solve the dynamic problems of a plate's free vibration. The method developed in ref. [4], on the other hand, is only suitable for a thin rectangular panel with combinations of classical boundaries. The method of Rayleigh–Ritz has then been introduced to study dynamical modeling for thin plates with non-classical boundaries. The Rayleigh–Ritz method is employed to study the dynamic properties of multi-point supported plates by Li et al. [5]. Dong [6] studied the free vibration of composite plates in three dimensions using the Chebyshev–Ritz method. The modal function constructed in this method always shows the complex product forms in terms of geometric boundary conditions, which slow down computation speed. Li et al. [7] used a set of beam functions based on the Rayleigh–Ritz method to investigate the dynamic properties of the folded solar arrays. Cao et al. [8] employed the same method to study the free vibrations of the flexible multi-panel structure. Although the research undertaken by Cao et al. was the first to investigate the analysis of the dynamic characteristics of flexible multi-panel structures, an important issue remains to be improved for its practical use: The free-free beam function consists of several trigonometric functions, which leads to the relatively slow computational speed and rate of convergence. The characteristic orthogonal polynomials were successfully utilized to investigate the dynamic characteristics of simple structures [9] and structures with multiple panels [10]; it is still a novel task to adopt it for the rigid–flexible coupling multi-body systems' natural property investigations. For the purposes of this paper, adopting the modal function-derived approach proposed by Bhat [11], the Rayleigh–Ritz method can be employed to analyze the natural frequencies and the analytical global mode functions for LSFS containing elastic connections.

Flexible spacecraft dynamics modeling is an important branch of dynamic research on flexible multibody systems. Large flexible spacecraft are often composed of beams, plates, trusses, and other simple hinged structures. Large rigid body movements of spacecraft, such as large-area rapid orbital maneuvers and agile attitude adjustment, can trigger strong vibrations of large flexible structures, showing the coupled dynamic characteristics of typical rigid body motion and structural vibration. For this kind of complex system, it is key to design the attitude and vibration controller and analyze the nonlinear dynamic characteristics of the system. The LSFS investigated in this research is shown in Figure 1. How to efficiently develop an accurate low-dimensional dynamic model is still a significant scientific issue that urgently needs to be addressed. Flexible solar arrays are distributed parameter systems, and their displacement would need to be discretized in order to derive the discrete dynamical models, which are widely used in the dynamic

analysis and vibration control of flexible space vehicles. The two most commonly used methods for doing this are the finite element method (FEM) [12,13] and the modal approach that uses mode shapes [14–16]. Shabana [16] developed the absolute nodal coordinate method to establish the dynamic model in three dimensions. Sahoo and Singh [17] studied the complicated dynamic problems of sandwich panels by a FEM model. The goal of Frikha and Zghal [18] was to study the dynamic behavior of composite shell structures reinforced with functionally graded carbon nanotubes, and a linear dual discrete director finite element model is used to develop the governing equations of motion. Jen [19] developed an improved substructure synthesis method to perform dynamic analysis of the structures. Hablani [20] employed the FEM to derive a clear mathematical model for a complex vehicle. It can be seen that the finite element method is a commonly used dynamic modeling method for complex space combinatorial structures such as flexible spacecraft, and the whole modal of the system can be easily obtained by this method. However, the result obtained by the finite element method is often in numerical form and it is difficult to obtain analytical expression, so it is disadvantageous for further research on nonlinear dynamics. In addition, modal synthesis is an effective method to obtain the global modes of complex systems. However, this method needs to obtain the modal information of each substructure and combine the modes of each substructure to obtain the global modes of the system. Both of them typically have a large number of degrees of freedom, and it is always difficult to obtain the discrete low-dimensional dynamical model, unfriendly to the controller design. Nowadays, with the development of computer software technology, multi-body dynamic engineering simulation software has become one of the common tools for the dynamic analysis of flexible multi-body systems. Large commercial software, represented by Adams, can now perform rigid–flexible coupling dynamics analysis, and simulation software, represented by the MATLAB and Simulink modules, has become a widely used analysis tool in the field of control engineering. For the study of spacecraft dynamics and control problems, two software simulations can be used, but the problem can only be solved in simpler engineering situations, and there are still some difficulties in the analysis of rigid–flexible coupling dynamics of more complex flexible multi-body systems. To overcome these drawbacks of the FEM model and software model, researchers use the modal method to derive the analytical dynamical model, and Hughes et al. [21] proposed the modal truncation for flexible spacecraft, which made the modal method possible. The structure of large spacecraft is complex and changeable, and there are many large flexible accessories. It will be difficult to continue to use the hybrid coordinate method to model the dynamics of such complex spacecraft. The process is complex, and its accuracy makes it difficult to meet the requirements. At this time, the appearance of the mode synthesis method has solved the dynamic modeling problem of complex flexible spacecraft very well. Global modes can be obtained by the synthesis of component modes [22], and the Craig–Bampton method is the most representative one [23]. The modes given by this method are approximate, however, and their expressions can be complicated. Pan and Liu [24] adopted the assumed modal method to establish a complex flexible multibody dynamic model for satellites and considered the thermal effects of the system. It was shown by Richard [25] that the admissible basis functions of the structures should be constructed based on the geometric boundary conditions. Nicolas et al. [26] studied the dynamic modeling and conducted an analysis of spacecraft with variable tilt of flexible appendages. Milad et al. [27] established the high-order rigid–flexible coupled structural system and designed the vibration controller. Most research, however, focuses on the study of the characteristics of a single structure such as a beam or a plate. There is a paucity of research investigating the analytical global rigid–flexible coupling mode for multibody structures. On the other hand, flexible oscillations have a great influence on the rigid movement of the system [28], so if we continue to adopt the constrained modes and ignore the coupling effect, the model will be imprecise. Hence, this research proposes an analytical method to obtain the global mode function and a low-DOF dynamic model to design the cooperative controller for the complex system.

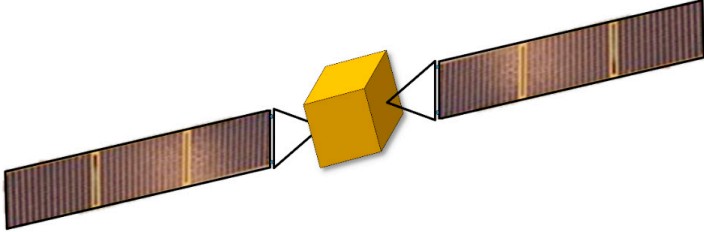

**Figure 1.** Diagram of the LSFS.

Due to the extreme rigid–flexible coupled effect of the LSFS, the vibration of flexible appendages is difficult to attenuate rapidly in space. It will have a great impact on the attitude accuracy of spacecraft. It is therefore of great importance to design an effective attitude–vibration cooperative controller. In the early research on the attitude control of spacecraft, the coupling of flexible vibration to body attitude motion was neglected, so the attitude control law of rigid spacecraft could not be applied directly to large flexible spacecraft. The design of modern large flexible spacecraft attitude and structure vibration controller should be based on a rigid–flexible coupling dynamic model of the system in order to control the attitude of spacecraft and suppress the vibration of flexible appendages. Classical control theory is suitable for single-input–single-output systems, and it is difficult to obtain a good control effect for multi-input–multi-output rigid–flexible coupled strong nonlinear systems with multi-control objectives and time variation. Modern spacecraft have many complex and varied tasks, and the classical theory of automatic control has been unable to meet the requirements of modern spacecraft. The emergence of modern control theory has been paid wide attention by many scholars, such as robust control, sliding structure control, and adaptive control. Although a lot of research studies have been conducted on spacecraft attitude movement and vibration control of flexible appendages, there are relatively few control strategies that can effectively suppress the vibration of flexible appendages while efficiently completing spacecraft attitude maneuvers.

Previous research ideas have been limited to equating the vibration of flexible appendages to an external disturbance, and then directly controlling the attitude. Another method is to design the controller completely separate from the attitude control and the vibration control of the spacecraft without considering the coupling effects. On the other hand, the vibration control of flexible spacecraft has also been experimentally studied on a flotation simulator by several scholars, and the experimental results were compared with the theoretical results of the proposed method. The simulator can realize the weightless movement of the experimental platform in the plane and the single-axis weightless rotation. A flexible L-beam is installed at the edge of the circular central platform. Two controllers, PD and bang-off-bang, were experimentally designed for the attitude control of spacecraft, and piezoelectric intelligent materials were used for the vibration control of flexible appendages. The experimental results show that the spacecraft can suppress the vibration of the flexible appendage and improve the efficiency and accuracy of the spacecraft's attitude maneuverability. The experimental results are in good agreement with the theoretical analysis. The problem of piezoelectric intelligent materials' actuator/sensor placement is also an issue itself but will not be addressed in this paper, and the detailed introduction about how to arrange the actuator position may be referred to in ref. [29]. Active vibration control such as strain rate feedback (SRF) control [30], positive position feedback (PPF) control [31], modal velocity feedback [32], linear quadratic regulator (LQR) control [33], optimal control [34], proportional-derivative (PD) control [35], and so on, are commonly used in engineering. An attractive feature of the SRF control approach is that the global stability of the entire system is guaranteed while the controllers accomplish their tasks in the coupled rigid–flexible dynamic domain without parasitic parameter interactions. The multi-modal vibration control of flexible structures is recognized as a significant challenge owing to the well-known phenomenon of spillover and frequency varying. The spillover phenomenon and variation in modal dynamics may cause instability

and degradation in performance. To overcome these defects, an adaptive positive position feedback (PPF) method is proposed for controlling the vibration of the system with bonded piezoelectric sensors and actuators. A multi-modal PPF controller can be adjusted with estimated frequencies and is able to damp the target modes quickly. A theoretical analysis of the flexural vibration of a system with a control system which implements direct velocity feedback using either an ideal collocated force actuator or a closely located piezoelectric patch actuator is usually presented to generate active damping which reduces the vibration level at resonance frequencies. Among them, a PD controller is easy to design and is widely used in industrial control. It has become one of the main control technologies because of its simple structure, good stability, reliable work, and convenient adjustment. When the system parameters of the controlled object cannot be fully grasped or an accurate mathematical model cannot be obtained, other technologies of control theory are difficult to adopt, and then the application of PD control technology is the most convenient. LQR is the earliest and most mature state space design method in modern control theory. The optimal solution of linear quadratic can be written into a unified analytical expression, and the solution process can be normalized. The state linear feedback control law can be simply used to form the closed-loop optimal control system, which can take into account multiple performance indexes. LQR optimal control can make the original system achieve better performance indexes by using a low cost, and the method is simple and easy to realize, which is convenient for realizing the stable, accurate, and fast control goals. In recent years, some scholars have proposed the strategy of combining input shaping (IS) with other closed-loop control methods. Different closed-loop control methods can be selected according to the characteristics of control tasks, and then various hybrid controllers with different emphases can be designed. Although the characteristics of various hybrid controllers are different, the core idea is to design a controller based on the better vibration suppression effect of the input shaping method and the strong anti-interference ability of the closed-loop control method. Its core idea is to divide a pulse into multiple sub-pulses, and the dynamic responses caused by multiple sub-pulses will just cancel each other out after the superposition, so as to achieve the purpose of vibration suppression. On account of the advantages of the three control methods, the LQR, PD, and PD + IS control methods are adopted to design a cooperative control scheme for the rigid–flexible coupled LSFS system.

In order to improve the dynamic modeling and cooperative control problems mentioned previously, a low-dimensional and high-precision dynamic model for the LSFS is established. The present method employs global analytical modes of the rigid–flexible coupling system to derive discrete governing equations by using the Hamiltonian principle. This paper investigates the attitude and vibration-coupled effects of in-orbit LSFS under three-axis attitude-driving torque. Furthermore, effective cooperative control schemes for attitude–vibration control are designed. The LQR, PD, and PD + IS controllers are designed based on the discrete dynamic model of the system. Through a typical example of spacecraft maneuvering from a known attitude angle to an expected attitude angle, the results of the LQR controller, the PD controller, and the PD + IS controller are compared, and the control effects and advantages of the three cooperative controllers designed in this paper are summarized. The cooperative controllers designed in our paper are based on the real rigid–flexible coupling global mode of the system, which can realize the attitude and vibration control synchronously, other than some designs of two independent controllers to reach the cooperative control goal as in most research studies. Our modeling approach can straightforwardly be applied to other multibody systems. Such an analytical solution and cooperative control for a model consisting of many panels and torsional springs have not been investigated in previous studies in the literature, and this manuscript seeks to fill this gap, which provides a general analytical method for the dynamic modeling and control for complex large-scale flexible spacecraft.

This paper is organized as follows: the procedures of description and discretization for the model of the LSFS are presented in Sections 2 and 3. The dynamic response analysis during the attitude maneuvering process is conducted in Section 4. The cooperative control

scheme for attitude motion and solar panel vibration control is designed in Section 5. Finally, some conclusions are summarized in the last section.

## 2. Mathematical Model of the LSFS

The dynamic modeling of flexible spacecraft is an important branch of flexible multibody system dynamics. Large flexible spacecraft are often assembled by hinged simple flexible structures such as beams, plates, trusses, etc. Large-scale rigid body motions of spacecraft, such as large-scale rapid orbit maneuvers, attitude agile adjustment, etc., may stimulate strong vibrations of large flexible structures, showing typical coupling dynamic characteristics of rigid body motion and structural vibration. For this complex system, how to accurately model and establish a mathematical model that can well describe its dynamic characteristics is the key to its attitude and vibration controller design and nonlinear dynamic characteristics analysis.

In order to establish an accurate and low-dimensional dynamic model of the LSFS, it is necessary to describe the detailed mathematical relationships and formulations for the whole system, including the assumptions, kinetic energy, and potential energy of the model.

### 2.1. Assumptions and Geometry Descriptions of the Model

The central body of the LSFS is rigid, and the solar panels are utilized as thin honeycomb plates. Each solar panel is connected by flexible hinges. Two yokes are used as rigid rods.

For the dynamic modeling of the hinges for the multi-panel structure, there are several common processing methods, as follows: use the finite element method to establish solid elements to directly model the hinge; the hinge is equivalent to a single beam structure for modeling; the hinge is simplified as a rotating joint with torsion spring for modeling; the connecting structure is simplified as a spring damping element; the hinge is modeled as a pendulum structure with spring constraints; and the mechanical model of the hinge is obtained by introducing contact theory. The equivalent beam method is one commonly used method to analyze the dynamic characteristics of the flexible solar array in the early days. It cannot accurately reflect the characteristics of the hinge when it works. Although it is simple to use, the calculation error is large. By using the equivalent spring method and directly measuring the hinge stiffness, a hinge model close to the actual situation can be obtained, so in this paper, flexible hinges are considered as rotational springs with no size, mass, damping, and Coulomb friction.

The honeycomb sandwich solar panels are mainly composed of upper and lower panels and the sandwich layer in the center. Generally, the panel and sandwich are bonded with adhesive to form a rigid structure. The honeycomb sandwich structure is widely used in aerospace and other fields for its light weight, low cost, high strength, sound insulation, and shock absorption, and it has become an indispensable structural material. In engineering applications, for complex structures such as the honeycomb sandwich, it is usually necessary to use the finite element method to analyze them, whereas the direct establishment of a detailed honeycomb model requires a large amount of calculation and requires huge time and cost. Therefore, in the calculation and analysis, it is often necessary to simplify it, so it is very important to establish a simple, accurate, and feasible equivalent model for the design and analysis of engineering structures. In order to simplify the honeycomb structure accurately, the research on equivalent parameters of the honeycomb core layer cannot be ignored. At present, most of the literature is based on the equivalent plate theory, the honeycomb plate theory, and the sandwich panel theory. The equivalent plate theory is that the whole honeycomb sandwich structure is equivalent to an isotropic plate with uniform density but different thicknesses. The theory of the honeycomb panel is that the whole honeycomb sandwich structure is equivalent to an orthotropic plate with the same stiffness and size, and the in-plane and out-of-plane mechanical properties of the faceplate and sandwich layer are considered simultaneously. The sandwich panel theory considers the panel and the sandwich layer separately. The panel is equivalent to

a homogeneous thin plate conforming to Kirchhoff's hypothesis, and the sandwich layer is equivalent to an orthotropic layer with a certain in-plane stiffness. We will not study the influence of the core honeycomb shape on the overall performance of the sandwich panel in this paper, so considering the equivalent plate theory is easy to realize, saving calculation cost and time. The panel of the solar array is equivalent to an isotropic elastic rectangular thin plate (shown in Figure 2) based on equivalent theory [36]. The equivalent material parameters $E_{eq}$, $\rho_{eq}$, $G_{eq}$, and $t_{eq}$ can be expressed as

$$
\begin{aligned}
t_{eq} &= \sqrt{12h_c^2 + 12h_c h_f + 4h_f^2}\,, & E_{eq} &= 2h_f E_f / t_{eq}\,, \\
G_{eq} &= 2h_f G_f / t_{eq}\,, & \rho_{eq} &= \left(2h_f \rho_f + 2h_c \rho_c\right)/t_{eq}\,,
\end{aligned}
\tag{1}
$$

where $\rho_c = \frac{8}{3}\frac{\delta_c}{l_c}\rho_0$ . $t_{eq} = H$.

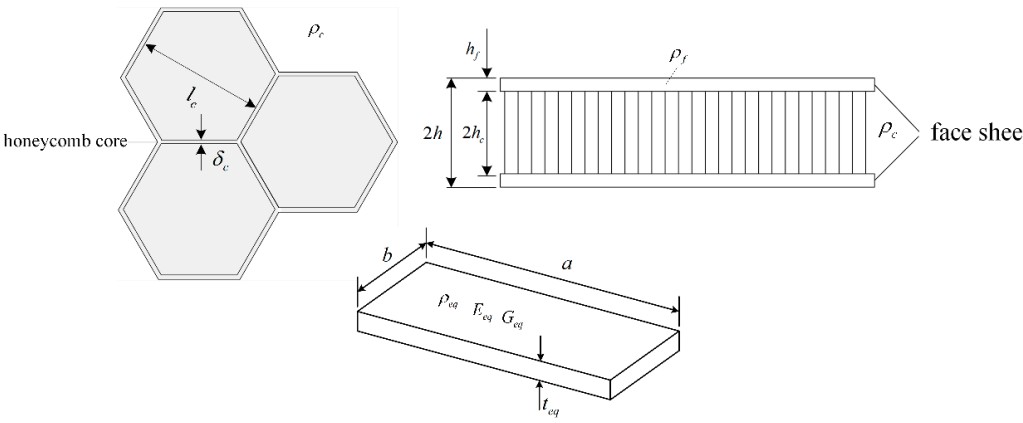

**Figure 2.** Equivalent isotropic model for the honeycomb sandwich panel.

The schematic diagram of the LSFS model and coordinate systems are defined in Figure 3. $O_0\text{-}x_0 y_0 z_0$ is the inertial reference coordinate system. $O\text{-}xyz$ is the floating coordinate system on the rigid cube.

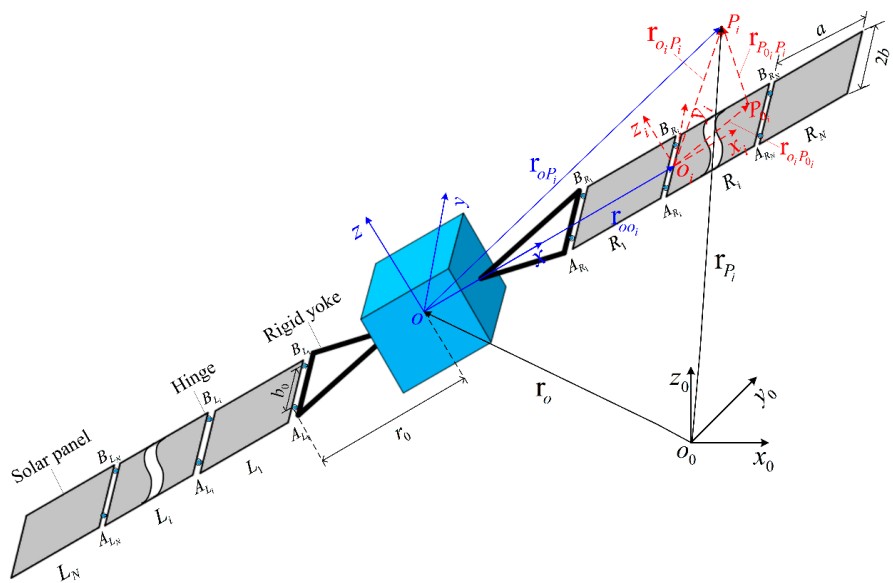

**Figure 3.** Model of the three-axis attitude stabilized LSFS.

The *O-xyz* to $O_0$-$x_0y_0z_0$ transformation matrix is

$$\mathbf{A}_{oo_0} = \begin{bmatrix} \cos\theta_z & -\sin\theta_z & 0 \\ \sin\theta_z & \cos\theta_z & 0 \\ 0 & 0 & 1 \end{bmatrix} \begin{bmatrix} 1 & 0 & 0 \\ 0 & \cos\theta_x & -\sin\theta_x \\ 0 & \sin\theta_x & \cos\theta_x \end{bmatrix} \begin{bmatrix} \cos\theta_y & 0 & \sin\theta_y \\ 0 & 1 & 0 \\ -\sin\theta_y & 0 & \cos\theta_y \end{bmatrix}. \tag{2}$$

$O_i$-$x_iy_iz_i$ is the floating frame on each plate. The $O_i$-$x_iy_iz_i$ to *O-xyz* transformation matrix is

$$\mathbf{A}_{o,o} = \begin{bmatrix} 1 & 0 & 0 \\ 0 & 1 & 0 \\ 0 & 0 & 1 \end{bmatrix}. \tag{3}$$

*2.2. The Expression of the LSFS's Solar Panel Displacement*

Each solar panel transverse displacement is given by

$$\begin{cases} w(x_{R_i}, y_{R_i}, t) = W(x_{R_i}, y_{R_i})\sin\omega t, \\ w(x_{L_i}, y_{L_i}, t) = W(x_{L_i}, y_{L_i})\sin\omega t, \end{cases} for\ i = 1, 2, \ldots, N. \tag{4}$$

where $\omega$ is the vibration frequency. $W(x_{R_i}, y_{R_i})$ and $W(x_{L_i}, y_{L_i})$ are the modal functions.

Based on the summary in the introduction section, the modal function $W(x_{R_i}, y_{R_i})$ and $W(x_{L_i}, y_{L_i})$ can be written as

$$W(x_{R_i}, y_{R_i}) = \sum_{m=1}^{m_t} \sum_{n=1}^{n_t} A_{mn}^{(R_i)} \varphi_m(x_{R_i}) \varphi_n(y_{R_i}),$$
$$W(x_{L_i}, y_{L_i}) = \sum_{m=1}^{m_t} \sum_{n=1}^{n_t} A_{mn}^{(L_i)} \varphi_m(x_{L_i}) \varphi_n(y_{L_i}). \tag{5}$$

where $\varphi_m(x_{R_i})$, $\varphi_n(y_{R_i})$, $\varphi_m(x_{L_i})$, and $\varphi_n(y_{L_i})$ are characteristic orthogonal polynomials in the *x* and *y* directions, respectively. $m_t$ and $n_t$ are numbers truncated when a specific model is calculated; $A_{mn}^{(R_i)}$ and $A_{mn}^{(L_i)}$ are the unknown coefficients.

*2.3. The LSFS's Kinetic Energy*

As illustrated in Figure 3, the position vector of an arbitrary point $P_i$ in $O_0$-$x_0y_0z_0$ can be expressed as

$$\mathbf{r}_{P_i} = \mathbf{r}_o + \mathbf{A}_{oo_0}\mathbf{r}_{oP_i} \tag{6}$$

Then, the velocity of the panel can be obtained as follows:

$$\mathbf{v}_{P_i} = \dot{\mathbf{r}}_{P_i} = \dot{\mathbf{r}}_o + \dot{\mathbf{A}}_{oo_0}\mathbf{r}_{oP_i} + \mathbf{A}_{oo_0}\dot{\mathbf{r}}_{oP_i} \tag{7}$$

Then, the kinetic energy of the whole system can be expressed as follows:

$$\begin{aligned} \mathrm{T} &= \frac{1}{2}\rho\sum_{i=1}^{N}\int_{V_{R_i}} \mathbf{v}_{P_{R_i}}^{\mathrm{T}} \cdot \mathbf{v}_{P_{R_i}} dV + \frac{1}{2}\rho\sum_{i=1}^{N}\int_{V_{L_i}} \mathbf{v}_{P_{L_i}}^{\mathrm{T}} \cdot \mathbf{v}_{P_{L_i}} dV \\ &+ \frac{1}{2}m_R(\dot{x}_o^2 + \dot{y}_o^2 + \dot{z}_o^2) + \frac{1}{2}\boldsymbol{\omega}^{\mathrm{T}} J_R \boldsymbol{\omega} \end{aligned} \tag{8}$$

where $\rho$ and $m_R$ are the density of each plate and the mass of the rigid platform. $x_o$, $y_o$, and $z_o$ are the positions of *O* in $O_0$-$x_0y_0z_0$. Additionally,

$$\boldsymbol{\omega} = \begin{bmatrix} \omega_x \\ \omega_y \\ \omega_z \end{bmatrix} = \begin{bmatrix} \cos\theta_y & 0 & -\cos\theta_x\sin\theta_y \\ 0 & 1 & \sin\theta_x \\ \sin\theta_y & 0 & \cos\theta_x\cos\theta_y \end{bmatrix} \begin{bmatrix} \dot{\theta}_x \\ \dot{\theta}_y \\ \dot{\theta}_z \end{bmatrix}$$

$$\mathbf{J_R} = \begin{bmatrix} J_x & 0 & 0 \\ 0 & J_y & 0 \\ 0 & 0 & J_z \end{bmatrix}$$

where $J_x$, $J_y$, and $J_z$ are the rigid hub inertial moments of the $x$-, $y$-, and $z$-axes.

Because the missions always need satellites to reach high-precision orientation, the satellite rotates very slowly, and the rotating angle is tiny when the attitude adjusts. Therefore, we are interested in whether the attitude angles will be small, so that Taylor expansion can be used to obtain the following first-order approximations for trigonometric functions of attitude angles, and the nonlinear terms are neglected here.

### 2.4. The LSFS's Potential Energy

Based on the relationship between the strain and stress of the panel, also taking into account the torsional spring, the potential energy of the LSFS can be expressed as

$$
\begin{aligned}
U \quad &= \sum_{i=1}^{N} \frac{D}{2} \int_{0}^{a} \int_{-b}^{b} \left[ \left( \frac{\partial^2 w_{R_i}}{\partial x_{R_i}^2} \right)^2 + 2v \frac{\partial^2 w_{R_i}}{\partial x_{R_i}^2} \frac{\partial^2 w_{R_i}}{\partial y_{R_i}^2} + \left( \frac{\partial^2 w_{R_i}}{\partial y_{R_i}^2} \right)^2 + 2(1-v) \left( \frac{\partial^2 w_{R_i}}{\partial x_{R_i} \partial y_{R_i}} \right)^2 \right] dx_{R_i} dy_{R_i} \\
&+ \sum_{i=1}^{N} \frac{D}{2} \int_{-a}^{0} \int_{-b}^{b} \left[ \left( \frac{\partial^2 w_{L_i}}{\partial x_{L_i}^2} \right)^2 + 2v \frac{\partial^2 w_{L_i}}{\partial x_{L_i}^2} \frac{\partial^2 w_{L_i}}{\partial y_{L_i}^2} + \left( \frac{\partial^2 w_{L_i}}{\partial y_{L_i}^2} \right)^2 + 2(1-v) \left( \frac{\partial^2 w_{L_i}}{\partial x_{L_i} \partial y_{L_i}} \right)^2 \right] dx_{L_i} dy_{L_i} \\
&+ \sum_{i=1}^{N} \frac{1}{2} k\Delta_{A_{R_i}}^2 + \sum_{i=1}^{N} \frac{1}{2} k\Delta_{B_{R_i}}^2 + \sum_{i=1}^{N} \frac{1}{2} k\Delta_{A_{L_i}}^2 + \sum_{i=1}^{N} \frac{1}{2} k\Delta_{B_{L_i}}^2
\end{aligned}
\tag{9}
$$

where $D$ denotes the flexural rigidity of the panel, $a$ and $b$ are the length and width of the panel, $k$ represents the stiffness of the rotational springs, and $\Delta_{A_{R_i}}$, $\Delta_{B_{R_i}}$, $\Delta_{A_{L_i}}$, and $\Delta_{B_{L_i}}$ denote the rotational angles of hinges $A_{R_i}$, $B_{R_i}$, $A_{L_i}$, and $B_{L_i}$ as shown in Figure 3, respectively.

## 3. Discrete Dynamic Model for the LSFS

The analytic global modes of the LSFS solved by the global mode method proposed in our previous research [37] are employed to conveniently establish an accurate discrete dynamic model here.

The $k$th order analytical global modes of the LSFS can be written as

$$
\boldsymbol{\Phi}_k = \left[ X_{o,k}, Y_{o,k}, Z_{o,k}, \theta_{0,k}^{(x)}, \theta_{0,k}^{(y)}, \theta_{0,k}^{(z)}, W_{R_1,k}, \dots, W_{R_N,k}, W_{L_1,k}, \dots, W_{L_N,k} \right]^{\mathrm{T}}
\tag{10}
$$

The displacement of the LSFS is given as follows:

$$
\begin{aligned}
& \left[ x_o, y_o, z_o, \theta_x, \theta_y, \theta_z, w_{R_1}, \dots, w_{R_N}, w_{L_1}, \dots, w_{L_N} \right]^{\mathrm{T}} \\
& = \left[ x_{or}, y_{or}, z_{or}, \theta_{xr}, \theta_{yr}, \theta_{zr}, \boldsymbol{0}_{1 \times 2N} \right]^{\mathrm{T}} + \boldsymbol{\Phi} \boldsymbol{p}(t)
\end{aligned}
\tag{11}
$$

where $\boldsymbol{p}(\mathrm{t})$ is the generalized modal coordinate vector expressed as

$$
\begin{cases}
\boldsymbol{\Phi} = [\boldsymbol{\Phi}_1, \boldsymbol{\Phi}_2, \dots, \boldsymbol{\Phi}_n] \\
\boldsymbol{p} = [p_1(t), p_2(t), \dots, p_n(t)]^{\mathrm{T}}
\end{cases}
\tag{12}
$$

Based on the parameter identification method, the linear transmitted torque formulation [38] of the $i$th hinge is given as

$$
M_s^{\mathrm{T}} = c\Delta\dot{\theta}_s + k\Delta\theta_s, \, (s = R_{A_i}, R_{B_i}, L_{A_i}, L_{B_i})
\tag{13}
$$

where the two terms of Equation (13) represent a linear damping and a linear spring, respectively. $c$ and $k$ are the linear damping coefficient and the linear spring stiffness coefficient, respectively.

Then, the damping is introduced in energy through Equation (13) and the Hamiltonian principle is employed to obtain the following discrete dynamic model:

$$
M\ddot{q} + C\dot{q} + Kq = Q
\tag{14}
$$

where *M*, *C*, and *K* are the mass, viscous damping, and stiffness matrices with the dimensions of $(6 + n) \times (6 + n)$, respectively. The details of the *M*, *C*, and *K* are given in the Appendix A. The *q* and *Q* are expressed as

$$q = \left[ x_{or}, y_{or}, z_{or}, \theta_{xr}, \theta_{yr}, \theta_{zr}, \boldsymbol{p}^T \right]^T$$
$$Q = \left[ 0, 0, 0, \tau_x, \tau_y, \tau_z, \tau_x \boldsymbol{\theta}_0^{(x)} + \tau_y \boldsymbol{\theta}_0^{(y)} + \tau_z \boldsymbol{\theta}_0^{(z)} \right]^T \tag{15}$$

## 4. The Validation Analysis for the Method

Based on the geometric and material properties of the LSFS studied in this paper, the finite element model of the system is established. Figure 4 is the finite element model of the LSFS in ANSYS, which is used to verify the accuracy and validity of the present method. The rigid hub is modeled by the Mass 21 element which has six DOFs, the mass, and the moment of inertia in the x, y, and z directions. Solar panels are discretized by employing the Shell 63 element. The Combine 14 element which has rotational stiffness is used to model the hinges.

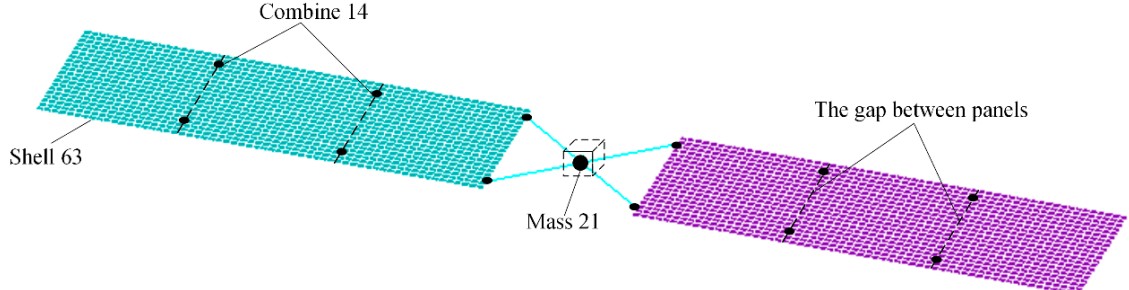

**Figure 4.** Finite element model of the LSFS in ANSYS.

The accuracy of the dynamic model is validated by comparing results obtained with the natural frequencies (taken as reference values) obtained from the finite element method in this subsection. As shown in Table 1, the first ten frequencies of the system calculated by commercial software ANSYS and the proposed method are given, respectively.

**Table 1.** Comparison of the first twelve frequencies of the system, *f* (Hz).

| Frequency Order | ANSYS | Proposed Method | $R_t$ (%) |
|:---:|:---:|:---:|:---:|
| 1 | 0.102 | 0.102 | 0.000 |
| 2 | 0.233 | 0.235 | 0.858 |
| 3 | 0.630 | 0.634 | 0.635 |
| 4 | 0.673 | 0.677 | 0.594 |
| 5 | 1.664 | 1.676 | 0.721 |
| 6 | 1.672 | 1.684 | 0.718 |
| 7 | 3.300 | 3.275 | 0.758 |
| 8 | 3.444 | 3.420 | 0.697 |
| 9 | 10.220 | 10.169 | 0.499 |
| 10 | 10.272 | 10.220 | 0.506 |

Here, the relative error is defined as

$$R_t = \left| \frac{f_{m_t n_t} - f_{\text{finite}}}{f_{\text{finite}}} \right| \times 100\% \tag{16}$$

where $f_{m_t n_t}$ is the frequency with respect to the polynomial terms of $m_t$ and $n_t$, and $f_{\text{finite}}$ represents the frequency calculated by ANSYS.

It can be observed from Table 1 that the absolute values of relative errors between the analytical solutions and the results from ANSYS are less than 0.858%. Hence, the accuracy of the model can be guaranteed by using the proposed method in this paper.

## 5. The Dynamic Response Analysis

The three-axis attitude-driving torque pulse $\boldsymbol{\tau}$ is

$$\boldsymbol{\tau} = \begin{bmatrix} \tau_x \\ \tau_y \\ \tau_z \end{bmatrix} = \begin{bmatrix} 0.1\tau_0 \\ \tau_0 \\ \tau_0 \end{bmatrix}, \tau_0 = \begin{cases} 10\text{N} \cdot \text{m}, & 0 \leq t \leq 4\text{s} \\ 0, & 4 < t < 8\text{s}, t > 12\text{s} \\ -10\text{N} \cdot \text{m}, & 8 \leq t \leq 12\text{s} \end{cases} \quad (17)$$

where $\tau_0$ is the amplitude of the attitude-driving torque pulse. The time history is shown in Figure 5.

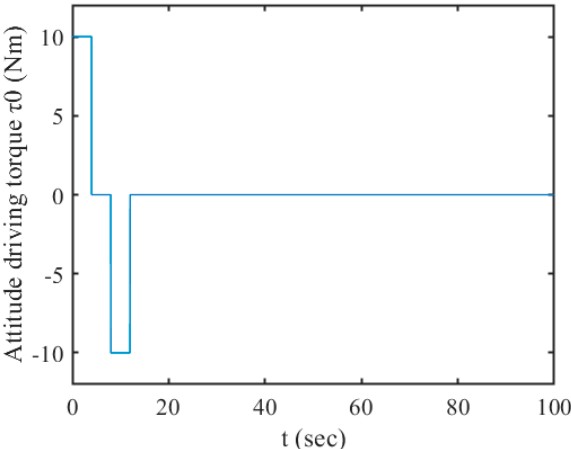

**Figure 5.** The attitude-driving torque pulse $\tau_0$.

The geometric and material properties of the LSFS studied in this section are listed in Table 2.

**Table 2.** Geometric and material parameters of the LSFS.

| Parameters | Values |
|---|---|
| The number of solar panels (N) | 3.0 |
| The length of the panel $a$ (m) | 2.0 |
| The width of the panel $2b$ (m) | 2.0 |
| Distance between the hinges $A$ and $B$ $b_0$ (m) | 1.6 |
| The thickness of the honeycomb core $2h_c$ (m) | 0.0197 |
| The thickness of the honeycomb face sheet $h_f$ (m) | $0.15 \times 10^{-3}$ |
| The length of the honeycomb wall $l_c$ (m) | $6.35 \times 10^{-3}$ |
| The thickness of the honeycomb wall $\delta_c$ (m) | $0.0254 \times 10^{-3}$ |
| The elastic modulus of the aluminum $E_0$ (Pa) | $6.89 \times 10^{10}$ |
| The mass density of the aluminum $\rho_0$ (kg m$^{-3}$) | $2.8 \times 10^3$ |
| Poisson ratio $v$ | 0.33 |
| The size of the distance $r_0$ (m) | 2.0 |
| The inertial moment of the hub $J_{x, y, z}$ (kg m$^2$) | 100,100,100 |
| The mass of the hub $m_R$ (kg) | 150 |
| The linear stiffness of the rotation spring $k$ (N · m/rad) | 50 |
| The damping coefficient of the hinges $c$ (N · m · s/rad) | 10 |

### 5.1. Displacements of the LSFS under the Three-Axis Attitude-Driving Torque Pulse

The rigid motions of the spacecraft in the $x$, $y$, and $z$ directions are shown in Figure 6. It can be observed that the vibration amplitudes in the $x$, $y$, and $z$ directions are almost

zero, which indicates the attitude maneuvering of the spacecraft can hardly excite any rigid translation motions, and it has almost no influence on the translational motions.

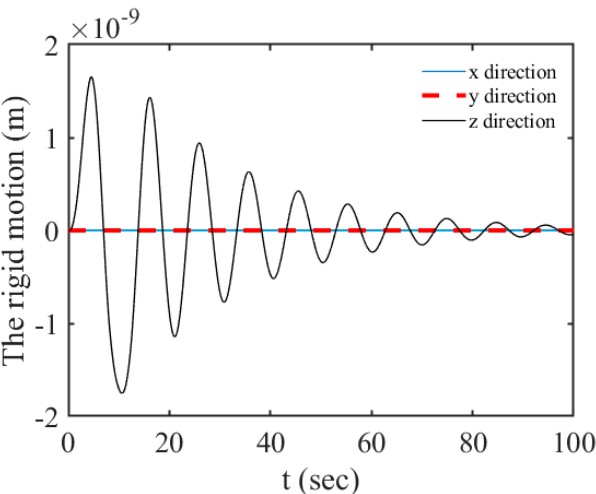

**Figure 6.** The translation of the spacecraft under the $\tau (k = 50\,\text{N} \cdot \text{m/rad}\,)$.

The attitude maneuvering angles of the spacecraft are plotted in Figure 7. It can be observed that the attitudes of the spacecraft have been adjusted to the expected positions gradually under the attitude-driving torque pulse. There is almost no fluctuation observed in Figure 7a,c. It is shown that the maneuvering process in the $\theta_x$ and $\theta_z$ directions cannot cause the vibration of the flexible solar arrays. However, in Figure 7b, the attitude angle $\theta_y$ oscillates remarkably, which demonstrates that the spacecraft attitude maneuvering in the $\theta_y$ direction will excite the rigid–flexible coupling effects. The vibrations of solar arrays also affect the attitude motions.

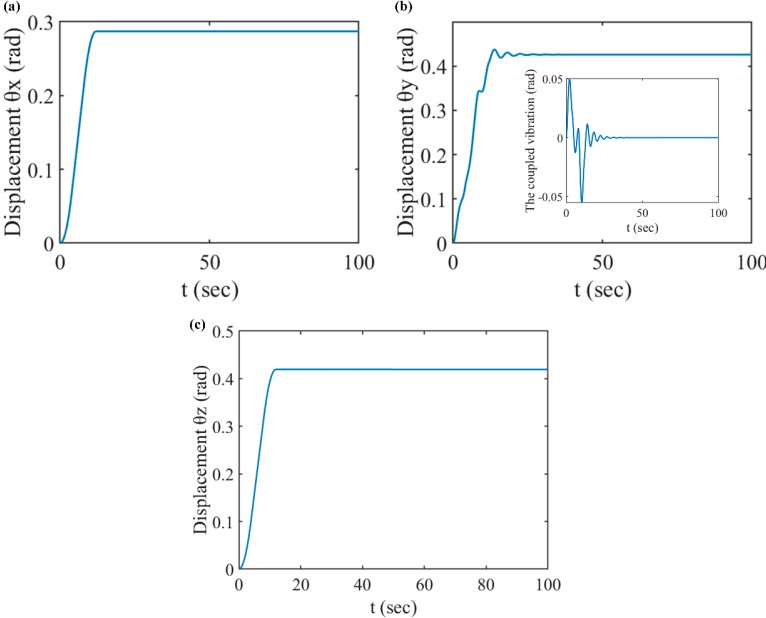

**Figure 7.** The attitude motion of the spacecraft under the $\tau (k = 50\,\text{N} \cdot \text{m/rad}\,)$; they are listed as: (**a**) the displacement of $\theta_x$; (**b**) the displacement of $\theta_y$; (**c**) the displacement of $\theta_z$.

Figure 8 shows the displacements of each of the right solar panels. The vibration amplitude of each panel increases gradually with the distance longer to the central body. In Figure 8, we can clearly observe that the displacement amplitudes of the solar panels jump exactly at 0 s, 4 s, 8 s, and 12 s, which means the sudden action and the sudden

stop of the attitude-driving torque can cause remarkable vibration of the solar array. The residual vibration lasted almost 30 s in Figure 8, and it was not conducive to the stability of spacecraft attitude. Therefore, designing an attitude–vibration cooperative controller is necessary.

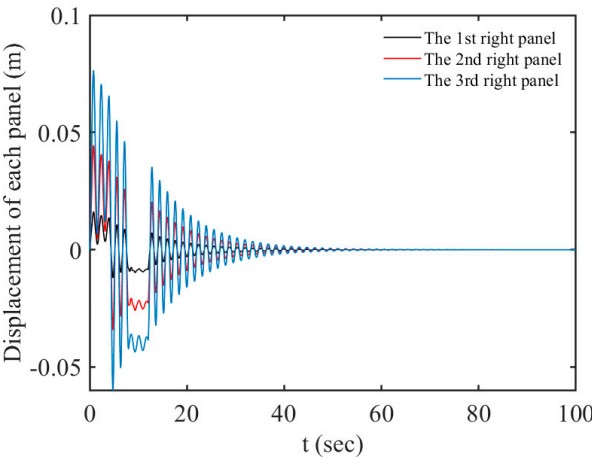

**Figure 8.** Displacements of the solar arrays under the $\tau (k = 50 \, \text{N} \cdot \text{m/rad})$.

## 5.2. The Effects of the Hinges for the Attitude Maneuver

In order to study the effects of the hinge flexure, the vibrations of the LSFS under various hinge stiffnesses are numerically carried out in the attitude maneuvering process.

The results in Figure 9a,d plainly show that as the stiffness $k$ increases, the vibration amplitudes of the solar array tips decrease. On the other hand, Figure 9 shows that increasing the stiffness of the hinge will extend the duration of the vibration and also will reduce the vibration periods. The flexibility of the hinge helps to accelerate the dissipation of the residual vibrations.

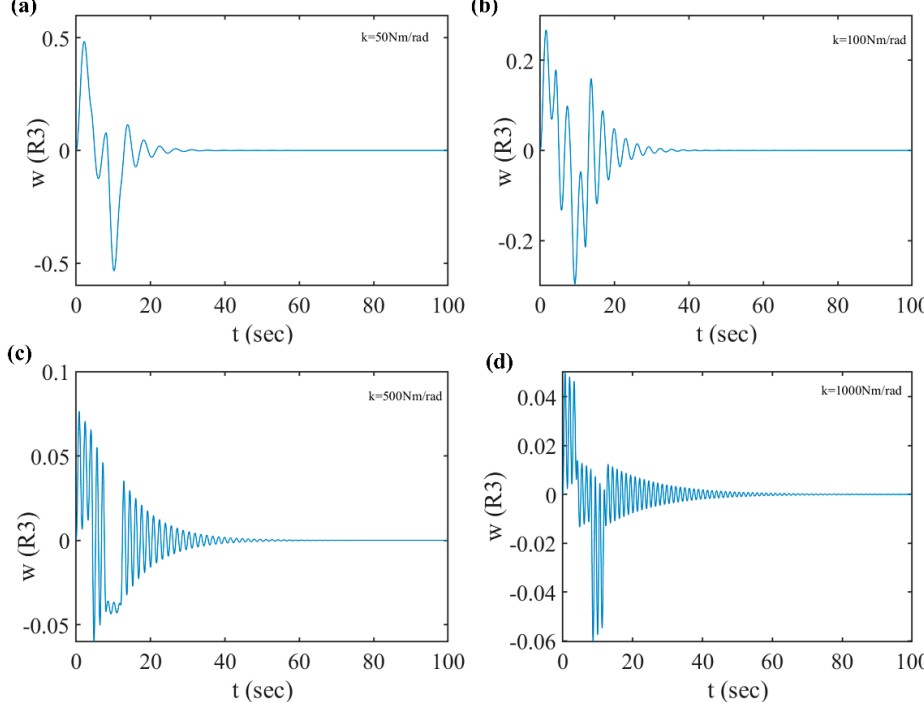

**Figure 9.** Deflections of the solar array tip for different spring stiffnesses; they are listed as: (**a**) $k = 50 \, \text{N} \cdot \text{m/rad}$; (**b**) $k = 100 \, \text{N} \cdot \text{m/rad}$; (**c**) $k = 500 \, \text{N} \cdot \text{m/rad}$; (**d**) $k = 1000 \, \text{N} \cdot \text{m/rad}$.

Figure 10 is plotted to discuss the effect of the hinge damping. It is illustrated that the different damping coefficients have a great influence on the fluctuations of the solar array. As can be seen from Figure 10, if the hinge damping is minimal, the residual vibration will persist for a long time. With the increase in the hinge damping, the vibration response of the solar array decreases, and the residual vibration dissipation time is shorter. Increasing the damping of hinges is an effective way to reduce the vibration response and accelerate the dissipation of residual vibration.

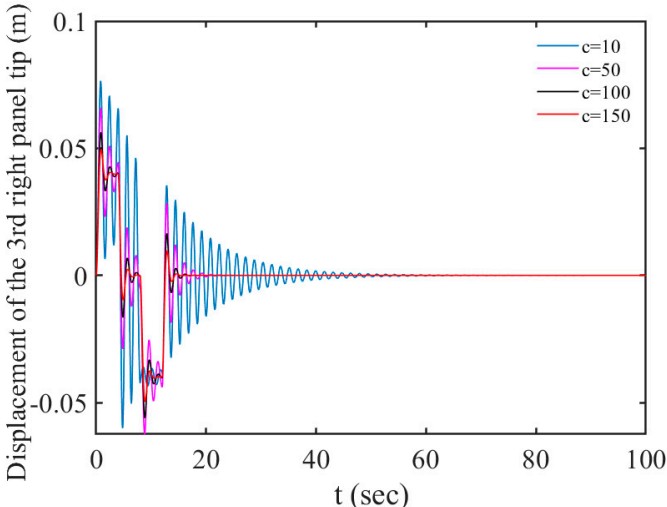

**Figure 10.** The vibration response of the system with various values of $c$ ($k = 500 \, \text{N} \cdot \text{m/rad}$).

## 6. The Cooperative Control for Attitude Motion and Structure Vibration

Based on the above analysis, we can conclude that only attitude motion $\theta_y$ excites the vibration of solar panels, causing the rigid–flexible coupling phenomenon. Therefore, the dimension of the model for the spacecraft considered in this paper can be reduced further by deleting the terms associated with other rigid motion variables. The degree of freedom of the new model is $1 + n$.

Taking the first few modes, the dynamic responses can be obtained by solving Equation (14). To determine how many modes should be taken for the controller design, the forced responses of the attitude motion $\theta_y$ and the end corner of the third-right panel with different numbers of modes are worked out through Equation (14) numerically, as shown in Figure 11a,b, respectively.

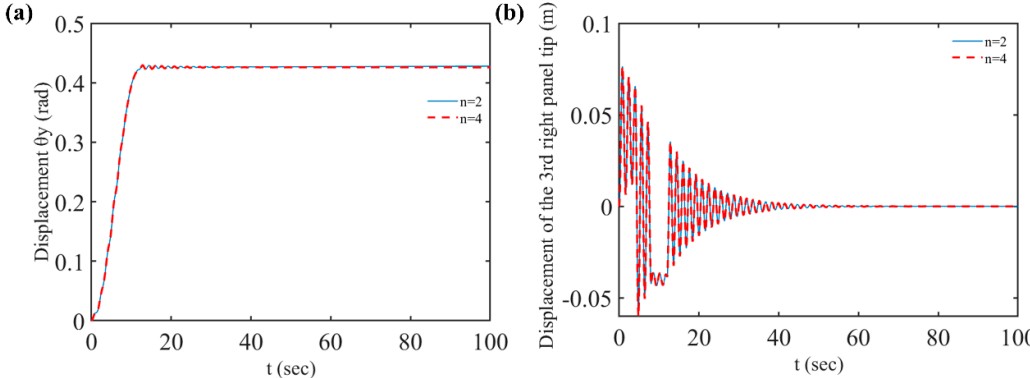

**Figure 11.** The vibration response of the system ($k = 500 \, \text{N} \cdot \text{m/rad}$); they are listed as: (**a**) the displacement of $\theta_y$; (**b**) the displacement of the 3rd-right panel tip.

It can be concluded from these figures that only the first two rigid–flexible coupling modes need to be selected to satisfy the accuracy of the dynamic responses analysis of the

system. From this point of view, the low-dimensional and high-precision dynamic model of the complex spacecraft can be obtained by using the global mode method, which can be used to design an accurate cooperative controller conveniently.

Then, two cooperative controllers are designed by employing the LQR and the PD control law for the attitude adjustment and vibration suppression of flexible spacecraft in this paper based on the three-DOFs model. As attitude movements $\theta_x$ and $\theta_z$ are decoupled from the flexible vibration of the solar array during the attitude adjustment process, only $\theta_y$ and $\tau_y$ are considered in the design of the cooperative controller. Here, force $Q$ in Equation (15) can be rewritten as follows:

$$Q = S\tau_y, \quad S = \left[1, \quad \left(\theta_0^{(y)}\right)_{1 \times n}\right]^{\mathrm{T}}. \tag{18}$$

*6.1. The LQR Cooperative Controller*

For the controller design problem of linear systems, if the performance index is the integral of the quadratic function of the state variables and (or) control variables, the optimization problem of such dynamic systems is called the optimal control problem of the quadratic performance index of linear systems, which is called the linear quadratic optimal control problem or the linear quadratic problem for short. The optimal solution of the linear quadratic problem can be written into a unified analytical expression, and the normalization of the solution process can be realized. The closed-loop optimal control system can be simply constructed by using the state linear feedback control law, which can take into account multiple performance indicators. Therefore, it has received special attention and is a more mature part of the modern control theory.

LQR (linear quadratic regulator) is a linear quadratic regulator. Its object is a linear system given in the form of state space in modern control theory, and the objective function is a quadratic function of object state and control input. LQR optimal design refers to the state feedback controller $\tau_y$ designed to minimize the quadratic objective function J, and $\tau_y$ is uniquely determined by the weight matrix Q and R, so the selection of Q and R is particularly important. LQR theory is the earliest and most mature state space design method in modern control theory. The fact that LQR can obtain the optimal control law of state linear feedback is particularly valuable, which is easy to use to form closed-loop optimal control. Moreover, the application of computer software provides conditions for LQR theoretical simulation, and it is more convenient for us to achieve stable, accurate, and fast control objectives.

In this section, the cooperative controller is designed by using the linear quadratic regulator (LQR) method. Figure 12 shows the block diagram of the LQR controller. The detailed procedures are demonstrated as follows:

$$\dot{Z} = AZ + B\tau_y. \tag{19}$$

where

$$Z = \begin{bmatrix} q \\ \dot{q} \end{bmatrix}, \quad A = \begin{bmatrix} 0 & I \\ -M^{-1}K & -M^{-1}C \end{bmatrix}, \quad B = \begin{bmatrix} 0 \\ M^{-1}S \end{bmatrix}.$$

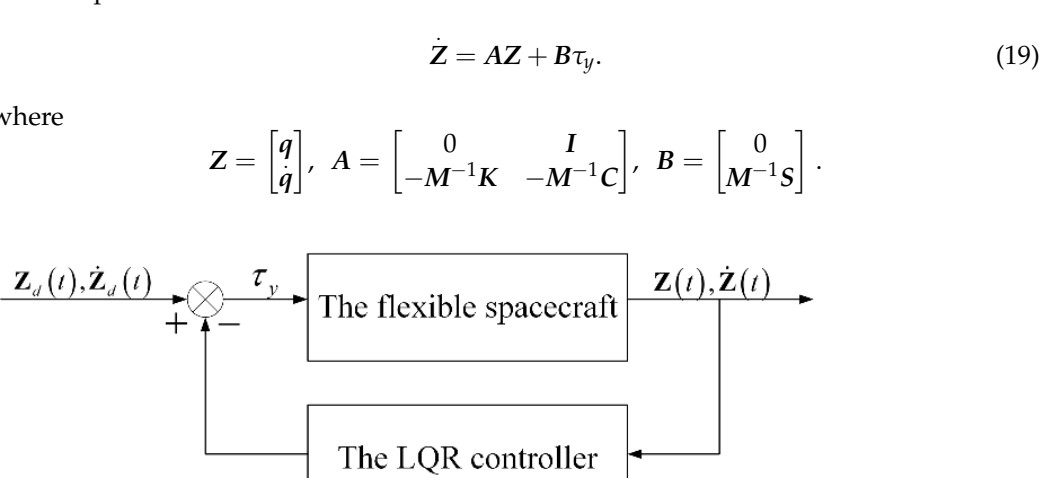

**Figure 12.** Block diagram of the LQR controller.

The control torque $\tau_y$ is defined as

$$\tau_y = -\boldsymbol{GZ},\tag{20}$$

where $\boldsymbol{G}$ is the control gain matrix. $\tau_y$ minimizes a quadratic performance index that is a cost function of the system states and control input.

$$J = \frac{1}{2}\int_0^\infty \left[\boldsymbol{Z}^{\mathrm{T}}\boldsymbol{Q}\boldsymbol{Z} + R\tau_y^2\right]dt,\tag{21}$$

where $\boldsymbol{Q}$ is a positive semimatrix, and $R$ is a positive weighting scalar. The gain matrix $\boldsymbol{G}$ can be obtained as $\boldsymbol{G} = R^{-1}\boldsymbol{B}^{\mathrm{T}}\boldsymbol{P}$, by solving the functional extremum problem, where $\boldsymbol{P}$ is the solution of the following Riccati equation:

$$\boldsymbol{PA} + \boldsymbol{A}^{\mathrm{T}}\boldsymbol{P} - \boldsymbol{PB}R^{-1}\boldsymbol{B}^{\mathrm{T}}\boldsymbol{P} + \boldsymbol{Q} = 0.\tag{22}$$

Here, the Riccati equation is the simplest nonlinear equation. *A*, *B*, *Q*, and *R* are known real coefficient matrices, and *P* is an unknown matrix. Generally, there are many solutions to this equation, but if there is a stable solution, we hope to find a stable solution. In the optimal control problem of infinite time, we focus on the value of some variables after a certain time, so we need to select the value of the control variable now so that the system can operate in the optimal state in the future. The optimal value of the control variable at any time can be obtained from the solution of the Riccati equation and the observed value of the state variable at that time. If there is more than one observation variable and control variable, the Riccati equation will be a matrix equation. The steady-state solution of *P* is related to the infinite time problem when approaching infinity. The dynamic equation can be iterated repeatedly until convergence to obtain the steady-state solution of *P*, and then the time label in the dynamic equation can be removed to confirm whether the steady-state solution is correct. If the algebraic Riccati equation has a stable solution, the solver will generally try to find a unique stable solution. The stable solution means that the closed-loop system can be stabilized by controlling the relevant LQR system with this solution.

The attitude sensor is installed on the central rigid body, and two displacement sensors are installed on the solar wing. The measured values of $\theta_y$, $\dot{\theta}_y$, $w$, and $\dot{w}$ can be obtained in real time. Then, the modal displacement and modal velocity of the system can be inversely calculated using Equation (14). Finally, the modal displacement and velocity are substituted into the attitude motion–structure vibration cooperative controller designed in this section to obtain the control torque of the controller.

### 6.2. The PD Cooperative Controller

As the most commonly used and easily realized control method in engineering, PD control still has the most extensive engineering applications. This kind of control method with high control accuracy and good dynamic performance is often used for three-axis stabilized spacecraft and does not require an accurate mathematical model. With the development of modern control theory, PD control has been constantly improved in practice. Self-tuning, adaptive, intelligent, and other improved types have emerged to adapt to various systems. The PD controller is proposed for flexible multibody systems, which are easy to implement and independent of spacecraft model parameters.

In this section, the cooperative controller is designed by using the PD control law. The block diagram of the PD controller is illustrated in Figure 13.

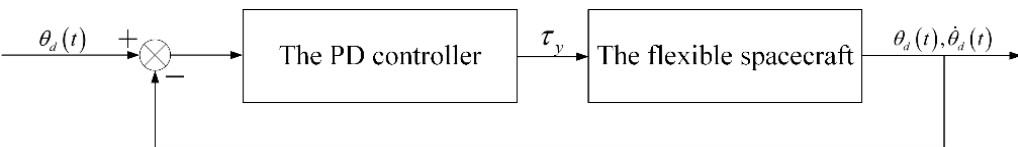

**Figure 13.** Block diagram of the PD controller.

The control torque $\tau_y$ is defined as

$$\tau_y = K_d\,\dot{e} + K_p\,e. \tag{23}$$

where tracking errors can be expressed as $\dot{e} = \dot{\theta}_d - \dot{\theta}$ and $e = \theta_d - \theta$. $K_d$ and $K_p$ are the differential gain and proportional gain, and $\theta_d$ and $\dot{\theta}_d$ are the desired attitude angle and the desired attitude angular velocity of the spacecraft, respectively. Substituting the control torque $\tau_y$ into Equation (23), the desired attitude angle of the flexible spacecraft can be accurately achieved, and the vibration of the solar arrays can be suppressed synchronously.

We installed an attitude sensor on the central rigid body, measured the attitude angle in real time, and substituted the error between the measured value and the expected value into the controller to obtain the control torque.

*6.3. The PD + IS Cooperative Controller*

In recent years, some scholars have proposed the strategy of combining input shaping (IS) with other closed-loop control methods. Different closed-loop control methods can be selected according to the characteristics of control tasks, and then various hybrid controllers with different emphases can be designed. Although the characteristics of various hybrid controllers are different, the core idea is to design the controller based on the better vibration suppression effect of the input shaping method and the strong anti-interference ability of the closed-loop control method. In 1958, Smith put forward the Posicast control method in his monograph "Feedback control systems", which was the first to put forward the idea of input shaping. Its core idea is to divide a pulse into multiple sub-pulses, and the dynamic responses caused by multiple sub-pulses will just cancel each other after superposition, so as to achieve the purpose of vibration suppression.

In this paper, the input shaping method and the proportional derivative (PD) control method are organically combined, and a PD + IS cooperative controller is designed for the flexible spacecraft, which realizes the simultaneous control of spacecraft attitude and structural vibration.

Here, $\theta_d$ is replaced by the pulse sequence given by the following formula:

$$\theta_{dIS} = \theta_d * A_{\text{mult}} \tag{24}$$

$A_{\text{mult}}$ is a multimodal input shaper with the following expression:

$$A_{\text{mult}} = A_{1s} * A_{2s} * \cdots * A_{is} * \cdots * A_{ns} \tag{25}$$

where $A_{is}$ ($i = 1, 2, \ldots, n$) is the pulse sequence of the $i$th mode of the system, including the pulse amplitude $A_j$, and the action time $t_j$; "$*$" is a convolution symbol. During calculation, the $A_j$ of each $A_{is}$ is multiplied, and the corresponding tj is added.

According to the design method of the ZVD shaper, the pulse amplitude and pulse time of the $j$th order frequency can be calculated from $\zeta$ and $\omega_d$:

$$\begin{bmatrix} A_j \\ t_j \end{bmatrix} = \begin{bmatrix} \frac{1}{(1+K)^2} & \frac{2K}{(1+K)^2} & \frac{K^2}{(1+K)^2} \\ T_1 & T_2 & T_3 \end{bmatrix}, \quad j = 1,\,2,\,3 \tag{26}$$

where $K = \exp(-\zeta\pi/\sqrt{1-\zeta^2})$, $T_j = (j-1)\pi/\omega_d$ ($j = 1,\,2,\ldots$). $\zeta$ and $\omega_d$ are the damping ratio and damped frequency of the $i$th order of the system, calculated by the following formula:

$$\lambda_{\text{sys}} = -\zeta\omega_d \pm i\omega_d\sqrt{1-\zeta^2} \tag{27}$$

If the newly generated pulse sequence is embedded into the feedback link in Figure 13, the error of the controller becomes $e = \theta_{dIS} - \theta$. In this way, the vibration of the solar array can be suppressed while the attitude is adjusted, forming a cooperative controller for attitude motion and solar array vibration suppression as shown in Figure 14.

**Figure 14.** Block diagram of the PD + IS controller.

In PD + IS control, the expected input instructions are obtained according to the task requirements of the system, so as to successfully complete the predetermined objectives of the system. The pulse sequence needs to be designed based on the dynamic characteristics of the system. The key is to obtain the accurate natural frequency and modal damping of the system. In this way, designing the desired input as multiple pulse sequences can ensure that the system can effectively suppress the flexible vibration while completing the set motion target.

*6.4. The Simulations and Discussions*

In this section, numerical simulations were conducted and presented to demonstrate the effectiveness of the control schemes designed in this paper. In this simulation, we tested the controllers against the complete distributed parameter model of the spacecraft, and the flexible spacecraft was commanded to a rest-to-rest maneuver, and the attitude angle varied from the initial state $\frac{\pi}{6}$ rad to the desired angle 0 rad. The desired attitude angular velocity equaled zero. The vibration of the spacecraft had to be suppressed when the attitude maneuver process finished.

For the LQR controller, the initial conditions of the system were chosen as follows: $t = 0$; $\theta_y = \frac{\pi}{6}$; $\dot{\theta}_y = 0$ rad s$^{-1}$; $p_i = 0$, $\dot{p}_i = 0$, $i = 1, 2$. The desired state variables $Z_d$ and $\dot{Z}_d$ were set to zero. Let $R = 1$, $Q = \text{diag}(5, 1, 1, 5000, 1, 1)$.

Figure 15 shows the simulation results for the LQR controller. It can be concluded from Figure 15a that the desired attitude angle of the flexible spacecraft can be accurately achieved within 100 s, and no overshoot occurs during attitude maneuvering. From Figure 15c, we can know the vibration of the hinge is not very strong during the attitude maneuvering process. However, at the beginning of the attitude maneuvering, the curves for the displacements of the solar panel $w_3$ and the hinge $B_{R_1}$ oscillate intensely. The relatively large amplitude vibration of the solar array is observed, and the maximum amplitude of $w_3$ reaches up to 8.8 mm. In addition, the maximum control torque is about 1.2 Nm.

For the PD controller, the attitude angle $\theta_d$ varies from the initial state $\theta_d = \frac{\pi}{6}$ rad to the desired angle 0 rad. The desired attitude angular velocity $\dot{\theta}_d$ equals zero. The PD gains of the PD controller are taken as $K_d = 48$ and $K_p = 1.2$, respectively.

The desired state of the flexible spacecraft for the PD controller remains the same as the LQR controller for a fair comparison, and the detailed simulation results are shown in Figure 16. It can be seen from Figure 16a that the time history curve of $\theta_y$ for the PD controller is essentially the same as with the LQR controller, and those two controllers have the capability to accomplish attitude stabilization within the same time. In Figure 16b,c, the curves for the displacements of the solar panel $w_3$ and hinge $B_{R_1}$ also oscillate intensely at the beginning of the attitude maneuvering. However, the maximum vibration amplitude of the solar array $w_3$ reaches up to 4.8 mm. In addition, the maximum control torque is about 0.7 Nm.

From the analyses above, compared to the LQR controller, the PD controller needs the smaller control torque to achieve the same attitude control goal for the spacecraft, and it causes smaller vibrations during the attitude-adjusting process. It is concluded that two cooperative controllers can accurately accomplish the attitude maneuvering and effectively suppress the vibration of the solar arrays as well as the corresponding residual vibration. By contrast, PD control algorithms are very simple and conducive to real-time spacecraft control. On the one hand, it is neither possible nor feasible to measure the full state in practice. Therefore, a controller that requires full-state feedback, such as a linear quadratic regulator (LQR) controller, may be limited to this factor. For PD control, no

full-state feedback is required; only attitude angles and angular velocity measured through the sensors on the spacecraft are needed.

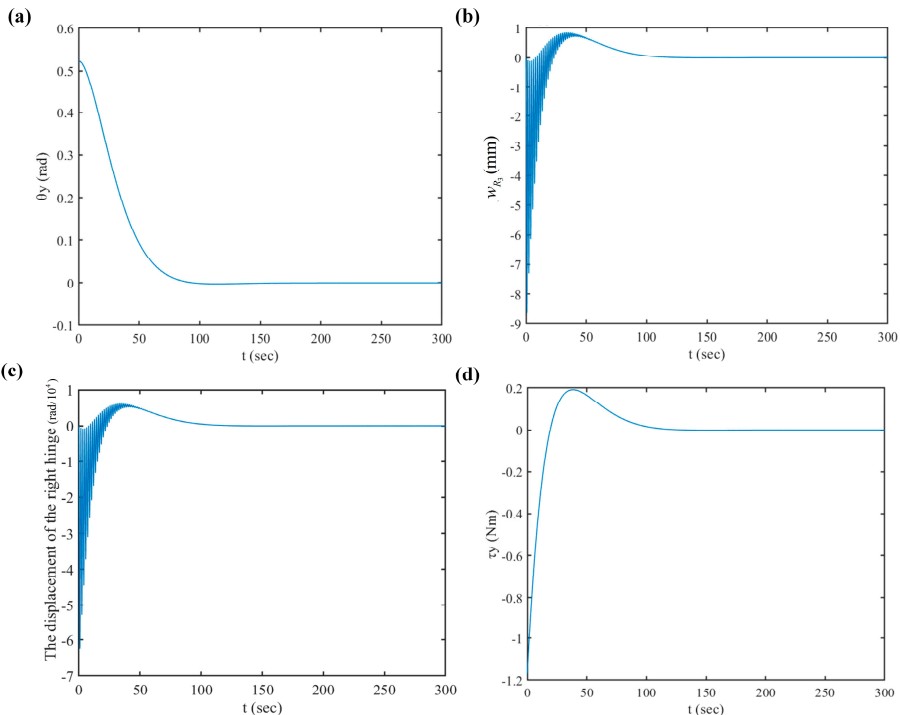

**Figure 15.** Time responses of the flexible spacecraft for using the LQR control ($k = 500$ N · m/rad); they are listed as: (**a**) the displacement of $\theta_y$; (**b**) the displacement of the 3rd-right panel tip; (**c**) the displacement of the right hinge $B_{R_1}$; (**d**) the control torque $\tau_y$.

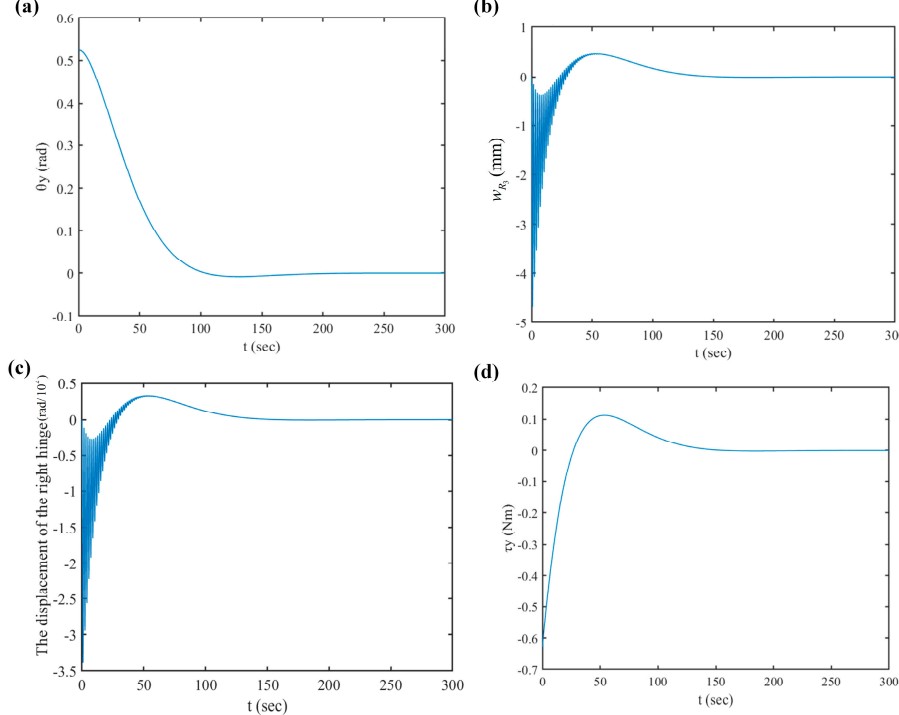

**Figure 16.** Time responses of the flexible spacecraft for using the PD control ($k = 500$ N · m/rad); they are listed as: (**a**) the displacement of $\theta_y$; (**b**) the displacement of the 3rd-right panel tip; (**c**) the displacement of the right hinge $B_{R_1}$; (**d**) the control torque $\tau_y$.

Under this condition, the expected attitude angle, angular velocity, and initial conditions of the spacecraft are the same as those of the PD controller simulation. The first two-order rigid flexible coupling global modes of the system are selected to design the multi-mode input shaper. The first and second-order modes are designed as ZVD shapers with three pulses. The simulation results are shown in Figure 17.

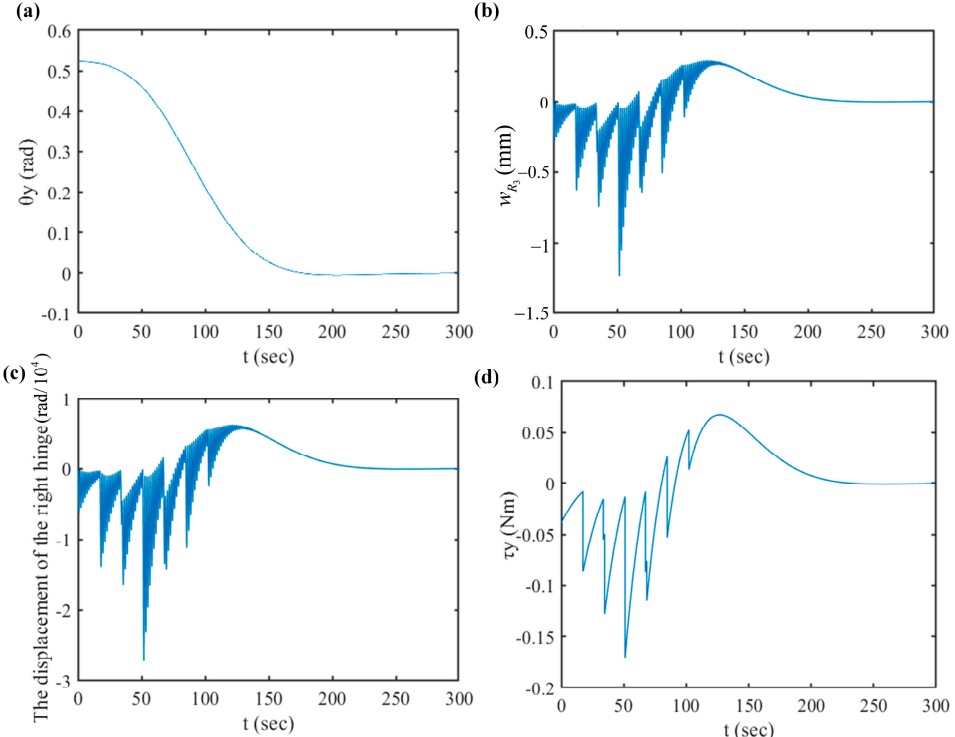

**Figure 17.** Time responses of the flexible spacecraft for using the PD + IS control ($k = 500 \, \text{N} \cdot \text{m/rad}$); they are listed as: (**a**) the displacement of $\theta_y$; (**b**) the displacement of the 3rd-right panel tip; (**c**) the displacement of the right hinge $B_{R_1}$; (**d**) the control torque $\tau_y$.

It can be seen from Figure 16a that the spacecraft accurately maneuvered to the expected attitude angle within 160 s without overshooting. The spacecraft solar array end vibration and hinge vibration in Figure 17b,c also gradually attenuated and disappeared after 160 s. Both of them have no residual vibration, which is equivalent to the effect of the PD controller. However, compared to the dynamic response under PD control in Figure 16, the PD + IS controller effectively suppressed the large amplitude oscillation of the flexible solar array excited by the attitude maneuvering at the initial stage of the control torque application. Through comparison, it can be seen that its amplitude decreased significantly. Under the control of PD + IS, the maximum value of the vibration displacement at the end of the solar array was 1.2 mm, far less than the 4.8 mm controlled by the PD. The control torque did not exceed 0.17 Nm, which is also smaller than that of the PD control. Through comparison, PD + IS control had a very significant effect on vibration suppression.

## 7. Conclusions

Considering a large-scale flexible spacecraft installed with a pair of multi-panel solar arrays, a novel approach is proposed to establish a discrete dynamic model with small dimensions and high precision for the system. The analytic global modes studied in our previous research studies are directly adopted for the discretization of the dynamic equations of flexible spacecraft and the design of a cooperative scheme for attitude control and vibration suppression.

The rigid–flexible coupling dynamic response of in-orbit LSFS under a three-axis attitude-driving torque pulse during the attitude maneuvering process are investigated. The

attitude maneuvering process of the flexible spacecraft can cause remarkable oscillations of its solar arrays. The hinges of the solar arrays have a significant effect on the dynamic characteristics of the spacecraft. This analysis shows that increasing the stiffness of the hinge is a method to reduce the vibration amplitude of the solar array, and the flexibility of the hinge can accelerate the elimination of residual vibrations. However, increased hinge stiffness was also observed to be detrimental to reducing vibration times. The discussion demonstrates that increasing hinge damping is an effective means of reducing the vibration response and accelerating the dissipation of residual vibration.

Moreover, the performance of cooperative controllers designed for attitude stabilization and vibration suppression based on the low-dimensional model is assessed by comparing them with controllers based on the LQR, PD, and PD + IS algorithms. The simulation results conducted on the distributed model of the system demonstrate that the controller designed by using the analytical low-dimensional model can suppress the oscillation of solar panels and accurately achieve the desired attitude angle within a short time. The LQR controller can achieve the effect of the cooperative control of spacecraft attitude and structural vibration, but it is a full-state feedback controller. In practical engineering, full-state measurement is neither possible nor feasible, so its application may be limited. Although the PD controller can also achieve the cooperative control effect of spacecraft attitude vibration, and it does not provide full-state feedback, which is easy to achieve, its vibration suppression effect is not as good as the PD + IS controller, so the PD + IS controller is an efficient attitude motion–structure vibration cooperative controller. It can not only accurately complete the spacecraft attitude maneuvering task but also effectively suppress the large amplitude and strong vibration of flexible components such as solar array substrates and hinges at the initial stage of spacecraft attitude-control torque. The PD + IS controller is not only simple to design but can also realize the real-time attitude structure vibration control of flexible spacecraft. However, the design of the PD + IS controller needs the precise dynamic characteristics of the system to be obtained so as to extract the cooperative accurate frequency and damping information, to design an input shaper suitable for the characteristics of the system. The global mode method presented in this paper can precisely obtain the precise modes of the complex system, providing a guarantee for the design of the attitude motion structure vibration cooperative controller of large flexible spacecraft. By comparison, the controller based on the PD + IS algorithm is very simple and thus more conducive to the real-time control of the spacecraft installed with a pair of solar arrays. Our approach can straightforwardly be applied to other multibody systems.

In this paper, the flexibility of the hinges for solar arrays was considered, and the interactions between the vibration and attitude of the spacecraft were studied. However, the nonlinear stiffness, the friction torque, the geometric nonlinearity of large deformation, the hinge clearance, and other nonlinear factors of the solar array were not considered. Therefore, it is necessary to further study these in the future. In a complex space environment, composite solar arrays will bear a time-varying thermal load to generate thermal flutter when a spacecraft enters and exits the Earth's shadow. Therefore, it is necessary to further establish the rigid flexible thermal coupling nonlinear dynamic model of attitude maneuver flexible spacecraft under solar heat flux and carry out research on the nonlinear dynamic characteristics of spacecraft under thermal load. The attitude motion and structure vibration cooperative controller based on the rigid flexible coupling global mode is designed, and a good control effect is achieved through simulation. However, the collaborative control laws designed in this paper have not been verified experimentally. Therefore, it is necessary to further carry out control experimental research on the basis of the theoretical research in this paper.

**Author Contributions:** Conceptualization, D.C.; methodology, G.H.; software, G.H.; validation, G.H. and D.C.; formal analysis, G.H.; investigation, G.H.; resources, D.C.; data curation, G.H.; writing— original draft preparation, G.H.; writing—review and editing, D.C.; visualization, G.H.; supervision, D.C.; project administration, D.C.; funding acquisition, D.C. All authors have read and agreed to the published version of the manuscript.

**Funding:** This research was funded by the National Natural Science Foundation of China, grant number 11732005 and the China Scholarship Council, grant number 202006120108.

**Data Availability Statement:** Not applicable.

**Acknowledgments:** The authors gratefully acknowledge the support of Dengqing Cao's research group, the Institute of Aerospace Vehicle Dynamics and Control, Harbin Institute of Technology.

**Conflicts of Interest:** The authors declare no conflict of interest.

## Appendix A

The mass matrix $\boldsymbol{M}$ is as follows:

$$\boldsymbol{M} = \begin{bmatrix} M_{11} & 0 & 0 & 0 & 0 & M_{16} & \boldsymbol{M}_{17} \\ 0 & M_{22} & 0 & 0 & 0 & M_{26} & \boldsymbol{M}_{27} \\ 0 & 0 & M_{33} & M_{34} & M_{35} & 0 & \boldsymbol{M}_{37} \\ 0 & 0 & M_{34}^{\mathrm{T}} & M_{44} & M_{45} & 0 & \boldsymbol{M}_{47} \\ 0 & 0 & M_{35}^{\mathrm{T}} & M_{45}^{\mathrm{T}} & M_{55} & 0 & \boldsymbol{M}_{57} \\ M_{16}^{\mathrm{T}} & M_{26}^{\mathrm{T}} & 0 & 0 & 0 & M_{66} & \boldsymbol{M}_{67} \\ \boldsymbol{M}_{17}^{\mathrm{T}} & \boldsymbol{M}_{27}^{\mathrm{T}} & \boldsymbol{M}_{37}^{\mathrm{T}} & \boldsymbol{M}_{47}^{\mathrm{T}} & \boldsymbol{M}_{57}^{\mathrm{T}} & \boldsymbol{M}_{67}^{\mathrm{T}} & \boldsymbol{M}_{77} \end{bmatrix}$$

where

$$M_{11} = 2N \cdot 4\rho hab + m_R$$

$$M_{16} = -2\rho h \sum_{i=1}^{N} \int_{0}^{a} \int_{-b}^{b} y_{R_i} \mathrm{d}y_{R_i} \mathrm{d}x_{R_i} - 2\rho h \sum_{i=1}^{N} \int_{-a}^{0} \int_{-b}^{b} y_{L_i} \mathrm{d}y_{L_i} \mathrm{d}x_{L_i}$$

$$M_{22} = 2N \cdot 4\rho hab + m_R$$

$$M_{26} = \rho \sum_{i=1}^{N} \int_{0}^{a} \int_{-b}^{b} 2h[x_{R_i} + r_0 + a(i-1)] \mathrm{d}y_{R_i} \mathrm{d}x_{R_i} + \rho \sum_{i=1}^{N} \int_{-a}^{0} \int_{-b}^{b} 2h[x_{L_i} - r_0 - a(i-1)] \mathrm{d}y_{L_i} \mathrm{d}x_{L_i}$$

$$M_{33} = 2N \cdot 4\rho hab + m_R$$

$$M_{34} = \sum_{i=1}^{N} \int_{0}^{a} \int_{-b}^{b} 2h y_{R_i} \mathrm{d}y_{R_i} \mathrm{d}x_{R_i} + \rho \sum_{i=1}^{N} \int_{-a}^{0} \int_{-b}^{b} 2h y_{L_i} \mathrm{d}y_{L_i} \mathrm{d}x_{L_i}$$

$$M_{35} = \rho \sum_{i=1}^{N} \int_{0}^{a} \int_{-b}^{b} -2h[x_{R_i} + r_0 + a(i-1)] \mathrm{d}y_{R_i} \mathrm{d}x_{R_i} + \rho \sum_{i=1}^{N} \int_{-a}^{0} \int_{-b}^{b} -2h[x_{L_i} - r_0 - a(i-1)] \mathrm{d}y_{L_i} \mathrm{d}x_{L_i}$$

$$M_{44} = \rho \sum_{i=1}^{N} \int_{0}^{a} \int_{-b}^{b} \frac{2}{3}h^3 + 2h y_{R_i}{}^2 \mathrm{d}y_{R_i} \mathrm{d}x_{R_i} + \rho \sum_{i=1}^{N} \int_{-a}^{0} \int_{-b}^{b} \frac{2}{3}h^3 + 2h y_{L_i}{}^2 \mathrm{d}y_{L_i} \mathrm{d}x_{L_i} + J_x$$

$$M_{45} = \rho \sum_{i=1}^{N} \int_{0}^{a} \int_{-b}^{b} -2h y_{R_i}[x_{R_i} + r_0 + a(i-1)] \mathrm{d}y_{R_i} \mathrm{d}x_{R_i} + \rho \sum_{i=1}^{N} \int_{-a}^{0} \int_{-b}^{b} -2h y_{L_i}[x_{L_i} - r_0 - a(i-1)] \mathrm{d}y_{L_i} \mathrm{d}x_{L_i}$$

$$\begin{aligned} M_{55} &= \rho \sum_{i=1}^{N} \int_{0}^{a} \int_{-b}^{b} \frac{2}{3}h^3 + 2h[x_{R_i} + r_0 + a(i-1)]^2 \mathrm{d}y_{R_i} \mathrm{d}x_{R_i} \\ &+ \rho \sum_{i=1}^{N} \int_{-a}^{0} \int_{-b}^{b} \frac{2}{3}h^3 + 2h[x_{L_i} - r_0 - a(i-1)]^2 \mathrm{d}y_{L_i} \mathrm{d}x_{L_i} + J_y \end{aligned}$$

$$
\begin{aligned}
M_{66} \quad &= \rho \sum_{i=1}^{N} \int_0^a \int_{-b}^b 2h \left\{ [x_{R_i} + r_0 + a(i-1)]^2 + y_{R_i}{}^2 \right\} \mathrm{d}y_{R_i} \mathrm{d}x_{R_i} \\
&+ \rho \sum_{i=1}^{N} \int_{-a}^0 \int_{-b}^b 2h \left\{ [x_{L_i} + r_0 + a(i-1)]^2 + y_{L_i}{}^2 \right\} \mathrm{d}y_{L_i} \mathrm{d}x_{L_i} + J_z
\end{aligned}
$$

$$
\begin{aligned}
\boldsymbol{M}_{17} \quad &= (2N \cdot 4\rho hab + m_R)\mathbf{X}_0 \\
&- \left( 2\rho h \sum_{i=1}^{N} \int_0^a \int_{-b}^b y_{R_i} \mathrm{d}y_{R_i} \mathrm{d}x_{R_i} + 2\rho h \sum_{i=1}^{N} \int_{-a}^0 \int_{-b}^b y_{L_i} \mathrm{d}y_{L_i} \mathrm{d}x_{L_i} \right) \theta_0^{(z)}
\end{aligned}
$$

$$
\begin{aligned}
\boldsymbol{M}_{27} \quad &= (2N \cdot 4\rho hab + m_R)\mathbf{Y}_0 + \left\{ \rho \sum_{i=1}^{N} \int_0^a \int_{-b}^b 2h[x_{R_i} + r_0 + a(i-1)] \, \mathrm{d}y_{R_i} \mathrm{d}x_{R_i} \right. \\
&\left. + \rho \sum_{i=1}^{N} \int_{-a}^0 \int_{-b}^b 2h[x_{L_i} - r_0 - a(i-1)] \, \mathrm{d}y_{L_i} \mathrm{d}x_{L_i} \right\} \theta_0^{(z)}
\end{aligned}
$$

$$
\begin{aligned}
\boldsymbol{M}_{37} \quad &= (2N \cdot 4\rho hab + m_R)\mathbf{Z}_0 \\
&+ \left( \rho \sum_{i=1}^{N} \int_0^a \int_{-b}^b 2hy_{R_i} \mathrm{d}y_{R_i} \mathrm{d}x_{R_i} + \rho \sum_{i=1}^{N} \int_{-a}^0 \int_{-b}^b 2hy_{L_i} \mathrm{d}y_{L_i} \mathrm{d}x_{L_i} \right) \theta_0^{(x)} \\
&- \left\{ \rho \sum_{i=1}^{N} \int_0^a \int_{-b}^b 2h[x_{R_i} + r_0 + a(i-1)] \, \mathrm{d}y_{R_i} \mathrm{d}x_{R_i} \right. \\
&\left. + \rho \sum_{i=1}^{N} \int_{-a}^0 \int_{-b}^b 2h[x_{L_i} - r_0 - a(i-1)] \mathrm{d}y_{L_i} \mathrm{d}x_{L_i} \right\} \theta_0^{(y)} \\
&+ \rho \sum_{i=1}^{N} \int_0^a \int_{-b}^b 2h\mathbf{W}_{R_i} \mathrm{d}y_{R_i} \mathrm{d}x_{R_i} + \rho \sum_{i=1}^{N} \int_{-a}^0 \int_{-b}^b 2h\mathbf{W}_{L_i} \mathrm{d}y_{L_i} \mathrm{d}x_{L_i}
\end{aligned}
$$

$$
\begin{aligned}
\boldsymbol{M}_{47} \quad &= \left( 2N \cdot \tfrac{4}{3}\rho hab^3 + 2N \cdot \tfrac{4}{3}\rho h^3 ab + J_x \right) \theta_0^{(x)} \\
&+ \left( \rho \sum_{i=1}^{N} \int_0^a \int_{-b}^b 2hy_{R_i} \mathrm{d}y_{R_i} \mathrm{d}x_{R_i} + \rho \sum_{i=1}^{N} \int_{-a}^0 \int_{-b}^b 2hy_{L_i} \mathrm{d}y_{L_i} \mathrm{d}x_{L_i} \right) \mathbf{Z}_0 \\
&- \left\{ \rho \sum_{i=1}^{N} \int_0^a \int_{-b}^b 2h[x_{R_i} + r_0 + a(i-1)] \, y_{R_i} \mathrm{d}y_{R_i} \mathrm{d}x_{R_i} \right. \\
&\left. + \rho \sum_{i=1}^{N} \int_{-a}^0 \int_{-b}^b 2h[x_{L_i} - r_0 - a(i-1)]y_{L_i} \mathrm{d}y_{L_i} \mathrm{d}x_{L_i} \right\} \theta_0^{(y)} \\
&+ \rho \sum_{i=1}^{N} \int_0^a \int_{-b}^b 2hy_{R_i}\mathbf{W}_{R_i} \mathrm{d}y_{R_i} \mathrm{d}x_{R_i} + \rho \sum_{i=1}^{N} \int_{-a}^0 \int_{-b}^b 2hy_{L_i}\mathbf{W}_{L_i} \mathrm{d}y_{L_i} \mathrm{d}x_{L_i} \\
&+ \rho \sum_{i=1}^{N} \int_0^a \int_{-b}^b \tfrac{2}{3}h^3 \frac{\partial \mathbf{W}_{R_i}}{\partial y_{R_i}} \mathrm{d}y_{R_i} \mathrm{d}x_{R_i} + \rho \sum_{i=1}^{N} \int_{-a}^0 \int_{-b}^b \tfrac{2}{3}h^3 \frac{\partial \mathbf{W}_{L_i}}{\partial y_{L_i}} \mathrm{d}y_{L_i} \mathrm{d}x_{L_i}
\end{aligned}
$$

$$
\begin{aligned}
\boldsymbol{M}_{57} \quad &= -\left\{ \rho \sum_{i=1}^{N} \int_0^a \int_{-b}^b 2h[x_{R_i} + r_0 + a(i-1)] \mathrm{d}y_{R_i} \mathrm{d}x_{R_i} \right. \\
&\left. + \rho \sum_{i=1}^{N} \int_{-a}^0 \int_{-b}^b 2h[x_{L_i} - r_0 - a(i-1)] \mathrm{d}y_{L_i} \mathrm{d}x_{L_i} \right\} \mathbf{Z}_0 \\
&- \left\{ \rho \sum_{i=1}^{N} \int_0^a \int_{-b}^b 2h[x_{R_i} + r_0 + a(i-1)] \, y_{R_i} \mathrm{d}y_{R_i} \mathrm{d}x_{R_i} \right. \\
&\left. + \rho \sum_{i=1}^{N} \int_{-a}^0 \int_{-b}^b 2h[x_{L_i} - r_0 - a(i-1)]y_{L_i} \mathrm{d}y_{L_i} \mathrm{d}x_{L_i} \right\} \theta_0^{(x)} \\
&+ \left\{ \rho \sum_{i=1}^{N} \int_0^a \int_{-b}^b 2h[x_{R_i} + r_0 + a(i-1)]^2 \mathrm{d}y_{R_i} \mathrm{d}x_{R_i} \right. \\
&\left. + \rho \sum_{i=1}^{N} \int_{-a}^0 \int_{-b}^b 2h[x_{L_i} - r_0 - a(i-1)]^2 \mathrm{d}y_{L_i} \mathrm{d}x_{L_i} + 2N \cdot \tfrac{4}{3}\rho h^3 ab + J_y \right\} \theta_0^{(y)} \\
&- \left\{ \rho \sum_{i=1}^{N} \int_0^a \int_{-b}^b 2h[x_{R_i} + r_0 + a(i-1)] \, \mathbf{W}_{R_i} \mathrm{d}y_{R_i} \mathrm{d}x_{R_i} \right. \\
&\left. + \rho \sum_{i=1}^{N} \int_{-a}^0 \int_{-b}^b 2h[x_{L_i} - r_0 - a(i-1)]\mathbf{W}_{L_i} \mathrm{d}y_{L_i} \mathrm{d}x_{L_i} \right\} \\
&+ \rho \sum_{i=1}^{N} \int_0^a \int_{-b}^b \tfrac{2}{3}h^3 \frac{\partial \mathbf{W}_{R_i}}{\partial x_{R_i}} \mathrm{d}y_{R_i} \mathrm{d}x_{R_i} + \rho \sum_{i=1}^{N} \int_{-a}^0 \int_{-b}^b \tfrac{2}{3}h^3 \frac{\partial \mathbf{W}_{L_i}}{\partial x_{L_i}} \mathrm{d}y_{L_i} \mathrm{d}x_{L_i}
\end{aligned}
$$

$$
\begin{aligned}
\boldsymbol{M}_{67} \;=&\; -\left(\rho\sum_{i=1}^{N}\int_{0}^{a}\int_{-b}^{b}2h\,y_{R_i}\mathrm{d}y_{R_i}\mathrm{d}x_{R_i}+\rho\sum_{i=1}^{N}\int_{-a}^{0}\int_{-b}^{b}2hy_{L_i}\mathrm{d}y_{L_i}\mathrm{d}x_{L_i}\right)\mathbf{X}_0 \\
&\;-\left\{\rho\sum_{i=1}^{N}\int_{0}^{a}\int_{-b}^{b}2h[x_{R_i}+r_0+a(i-1)]\mathrm{d}y_{R_i}\mathrm{d}x_{R_i}\right. \\
&\;\left.+\rho\sum_{i=1}^{N}\int_{-a}^{0}\int_{-b}^{b}2h[x_{L_i}-r_0-a(i-1)]\mathrm{d}y_{L_i}\mathrm{d}x_{L_i}\right\}\mathbf{Y}_0 \\
&\;+\left\{\rho\sum_{i=1}^{N}\int_{0}^{a}\int_{-b}^{b}2h[x_{R_i}+r_0+a(i-1)]^2\mathrm{d}y_{R_i}\mathrm{d}x_{R_i}\right. \\
&\;\left.+\rho\sum_{i=1}^{N}\int_{-a}^{0}\int_{-b}^{b}2h[x_{L_i}-r_0-a(i-1)]^2\mathrm{d}y_{L_i}\mathrm{d}x_{L_i}+2N\cdot\tfrac{4}{3}\rho hab^3+J_z\right\}\boldsymbol{\theta}_0^{(z)}
\end{aligned}
$$

$$
\begin{aligned}
\boldsymbol{M}_{77} \;=&\; M_{11}\mathbf{X}_0^{\mathrm{T}}\mathbf{X}_0+M_{22}\mathbf{Y}_0^{\mathrm{T}}\mathbf{Y}_0+M_{33}\mathbf{Z}_0^{\mathrm{T}}\mathbf{Z}_0+M_{44}\boldsymbol{\theta}_0^{(x)^{\mathrm{T}}}\boldsymbol{\theta}_0^{(x)}+M_{55}\boldsymbol{\theta}_0^{(y)^{\mathrm{T}}}\boldsymbol{\theta}_0^{(y)}+M_{66}\boldsymbol{\theta}_0^{(z)^{\mathrm{T}}}\boldsymbol{\theta}_0^{(z)} \\
&\;+\rho\sum_{i=1}^{N}\int_{0}^{a}\int_{-b}^{b}\tfrac{2}{3}h^3\frac{\partial\mathbf{W}_{R_i}^{\mathrm{T}}}{\partial x_{R_i}}\frac{\partial\mathbf{W}_{R_i}}{\partial x_{R_i}}\mathrm{d}y_{R_i}\mathrm{d}x_{R_i}+\rho\sum_{i=1}^{N}\int_{-a}^{0}\int_{-b}^{b}\tfrac{2}{3}h^3\frac{\partial\mathbf{W}_{L_i}^{\mathrm{T}}}{\partial x_{L_i}}\frac{\partial\mathbf{w}_{L_i}}{\partial x_{L_i}}\mathrm{d}y_{L_i}\mathrm{d}x_{L_i} \\
&\;+\rho\sum_{i=1}^{N}\int_{0}^{a}\int_{-b}^{b}\tfrac{2}{3}h^3\frac{\partial\mathbf{W}_{R_i}^{\mathrm{T}}}{\partial y_{R_i}}\frac{\partial\mathbf{W}_{R_i}}{\partial y_{R_i}}\mathrm{d}y_{R_i}\mathrm{d}x_{R_i}+\rho\sum_{i=1}^{N}\int_{-a}^{0}\int_{-b}^{b}\tfrac{2}{3}h^3\frac{\partial\mathbf{W}_{L_i}^{\mathrm{T}}}{\partial y_{L_i}}\frac{\partial\mathbf{w}_{L_i}}{\partial y_{L_i}}\mathrm{d}y_{L_i}\mathrm{d}x_{L_i} \\
&\;+\rho\sum_{i=1}^{N}\int_{0}^{a}\int_{-b}^{b}2h\mathbf{W}_{R_i}^{\mathrm{T}}\mathbf{W}_{R_i}\mathrm{d}y_{R_i}\mathrm{d}x_{R_i}+\rho\sum_{i=1}^{N}\int_{-a}^{0}\int_{-b}^{b}2h\mathbf{W}_{L_i}^{\mathrm{T}}\mathbf{W}_{L_i}\mathrm{d}y_{L_i}\mathrm{d}x_{L_i} \\
&\;+\rho\sum_{i=1}^{N}\int_{0}^{a}\int_{-b}^{b}2h[x_{R_i}+r_0+a(i-1)]\mathrm{d}y_{R_i}\mathrm{d}x_{R_i}\left(\boldsymbol{\theta}_0^{(z)^{\mathrm{T}}}\mathbf{Y}_0+\mathbf{Y}_0^{\mathrm{T}}\boldsymbol{\theta}_0^{(z)}\right) \\
&\;+\rho\sum_{i=1}^{N}\int_{-a}^{0}\int_{-b}^{b}2h[x_{L_i}-r_0-a(i-1)]\mathrm{d}y_{L_i}\mathrm{d}x_{L_i}\left(\boldsymbol{\theta}_0^{(z)^{\mathrm{T}}}\mathbf{Y}_0+\mathbf{Y}_0^{\mathrm{T}}\boldsymbol{\theta}_0^{(z)}\right) \\
&\;+\rho\sum_{i=1}^{N}\int_{0}^{a}\int_{-b}^{b}2h\left(\mathbf{Z}_0^{\mathrm{T}}\mathbf{W}_{R_i}+\mathbf{W}_{R_i}^{\mathrm{T}}\mathbf{Z}_0\right)\mathrm{d}y_{R_i}\mathrm{d}x_{R_i}+\rho\sum_{i=1}^{N}\int_{-a}^{0}\int_{-b}^{b}2h\left(\mathbf{Z}_0^{\mathrm{T}}\mathbf{W}_{L_i}+\mathbf{W}_{L_i}^{\mathrm{T}}\mathbf{Z}_0\right)\mathrm{d}y_{L_i}\mathrm{d}x_{L_i} \\
&\;-\rho\sum_{i=1}^{N}\int_{0}^{a}\int_{-b}^{b}2h[x_{R_i}+r_0+a(i-1)]\mathrm{d}y_{R_i}\mathrm{d}x_{R_i}\left(\mathbf{Z}_0^{\mathrm{T}}\boldsymbol{\theta}_0^{(y)}+\boldsymbol{\theta}_0^{(y)^{\mathrm{T}}}\mathbf{Z}_0\right) \\
&\;-\rho\sum_{i=1}^{N}\int_{-a}^{0}\int_{-b}^{b}2h[x_{L_i}-r_0-a(i-1)]\mathrm{d}y_{L_i}\mathrm{d}x_{L_i}\left(\mathbf{Z}_0^{\mathrm{T}}\boldsymbol{\theta}_0^{(y)}+\boldsymbol{\theta}_0^{(y)^{\mathrm{T}}}\mathbf{Z}_0\right) \\
&\;+\rho\sum_{i=1}^{N}\int_{0}^{a}\int_{-b}^{b}2hy_{R_i}\mathrm{d}y_{R_i}\mathrm{d}x_{R_i}\left(\mathbf{Z}_0^{\mathrm{T}}\boldsymbol{\theta}_0^{(x)}+\boldsymbol{\theta}_0^{(x)^{\mathrm{T}}}\mathbf{Z}_0\right)+\rho\sum_{i=1}^{N}\int_{-a}^{0}\int_{-b}^{b}2hy_{L_i}\mathrm{d}y_{L_i}\mathrm{d}x_{L_i}\left(\mathbf{Z}_0^{\mathrm{T}}\boldsymbol{\theta}_0^{(x)}+\boldsymbol{\theta}_0^{(x)^{\mathrm{T}}}\mathbf{Z}_0\right) \\
&\;-\rho\sum_{i=1}^{N}\int_{0}^{a}\int_{-b}^{b}2h[x_{R_i}+r_0+a(i-1)]y_{R_i}\mathrm{d}y_{R_i}\mathrm{d}x_{R_i}\left(\boldsymbol{\theta}_0^{(x)^{\mathrm{T}}}\boldsymbol{\theta}_0^{(y)}+\boldsymbol{\theta}_0^{(y)^{\mathrm{T}}}\boldsymbol{\theta}_0^{(x)}\right) \\
&\;-\rho\sum_{i=1}^{N}\int_{-a}^{0}\int_{-b}^{b}2h[x_{L_i}-r_0-a(i-1)]y_{L_i}\mathrm{d}y_{L_i}\mathrm{d}x_{L_i}\left(\boldsymbol{\theta}_0^{(x)^{\mathrm{T}}}\boldsymbol{\theta}_0^{(y)}+\boldsymbol{\theta}_0^{(y)^{\mathrm{T}}}\boldsymbol{\theta}_0^{(x)}\right) \\
&\;-\rho\sum_{i=1}^{N}\int_{0}^{a}\int_{-b}^{b}2hy_{R_i}\mathrm{d}y_{R_i}\mathrm{d}x_{R_i}\left(\boldsymbol{\theta}_0^{(z)^{\mathrm{T}}}\mathbf{X}_0+\mathbf{X}_0^{\mathrm{T}}\boldsymbol{\theta}_0^{(z)}\right)-\rho\sum_{i=1}^{N}\int_{-a}^{0}\int_{-b}^{b}2hy_{L_i}\mathrm{d}y_{L_i}\mathrm{d}x_{L_i}\left(\boldsymbol{\theta}_0^{(z)^{\mathrm{T}}}\mathbf{X}_0+\mathbf{X}_0^{\mathrm{T}}\boldsymbol{\theta}_0^{(z)}\right) \\
&\;-\rho\sum_{i=1}^{N}\int_{0}^{a}\int_{-b}^{b}2h[x_{R_i}+r_0+a(i-1)]\left(\mathbf{W}_{R_i}^{\mathrm{T}}\boldsymbol{\theta}_0^{(y)}+\boldsymbol{\theta}_0^{(y)^{\mathrm{T}}}\mathbf{W}_{R_i}\right)\mathrm{d}y_{R_i}\mathrm{d}x_{R_i} \\
&\;-\rho\sum_{i=1}^{N}\int_{-a}^{0}\int_{-b}^{b}2h[x_{L_i}-r_0-a(i-1)]\left(\mathbf{W}_{L_i}^{\mathrm{T}}\boldsymbol{\theta}_0^{(y)}+\boldsymbol{\theta}_0^{(y)^{\mathrm{T}}}\mathbf{W}_{L_i}\right)\mathrm{d}y_{L_i}\mathrm{d}x_{L_i} \\
&\;+\rho\sum_{i=1}^{N}\int_{0}^{a}\int_{-b}^{b}2hy_{R_i}\left(\mathbf{W}_{R_i}^{\mathrm{T}}\boldsymbol{\theta}_0^{(x)}+\boldsymbol{\theta}_0^{(x)^{\mathrm{T}}}\mathbf{W}_{R_i}\right)\mathrm{d}y_{R_i}\mathrm{d}x_{R_i} \\
&\;+\rho\sum_{i=1}^{N}\int_{-a}^{0}\int_{-b}^{b}2hy_{L_i}\left(\mathbf{W}_{L_i}^{\mathrm{T}}\boldsymbol{\theta}_0^{(x)}+\boldsymbol{\theta}_0^{(x)^{\mathrm{T}}}\mathbf{W}_{L_i}\right)\mathrm{d}y_{L_i}\mathrm{d}x_{L_i} \\
&\;-\rho\sum_{i=1}^{N}\int_{0}^{a}\int_{-b}^{b}\tfrac{2}{3}h^3\left(\frac{\partial\mathbf{W}_{R_i}^{\mathrm{T}}}{\partial x_{R_i}}\boldsymbol{\theta}_0^{(y)}+\boldsymbol{\theta}_0^{(y)^{\mathrm{T}}}\frac{\partial\mathbf{W}_{R_i}}{\partial x_{R_i}}\right)\mathrm{d}y_{R_i}\mathrm{d}x_{R_i} \\
&\;-\rho\sum_{i=1}^{N}\int_{-a}^{0}\int_{-b}^{b}\tfrac{2}{3}h^3\left(\frac{\partial\mathbf{W}_{L_i}^{\mathrm{T}}}{\partial x_{L_i}}\boldsymbol{\theta}_0^{(y)}+\boldsymbol{\theta}_0^{(y)^{\mathrm{T}}}\frac{\partial\mathbf{W}_{L_i}}{\partial x_{L_i}}\right)\mathrm{d}y_{L_i}\mathrm{d}x_{L_i} \\
&\;+\rho\sum_{i=1}^{N}\int_{0}^{a}\int_{-b}^{b}\tfrac{2}{3}h^3\left(\frac{\partial\mathbf{W}_{R_i}^{\mathrm{T}}}{\partial y_{R_i}}\boldsymbol{\theta}_0^{(x)}+\boldsymbol{\theta}_0^{(x)^{\mathrm{T}}}\frac{\partial\mathbf{W}_{R_i}}{\partial y_{R_i}}\right)\mathrm{d}y_{R_i}\mathrm{d}x_{R_i} \\
&\;+\rho\sum_{i=1}^{N}\int_{-a}^{0}\int_{-b}^{b}\tfrac{2}{3}h^3\left(\frac{\partial\mathbf{W}_{L_i}^{\mathrm{T}}}{\partial y_{L_i}}\boldsymbol{\theta}_0^{(x)}+\boldsymbol{\theta}_0^{(x)^{\mathrm{T}}}\frac{\partial\mathbf{W}_{L_i}}{\partial y_{L_i}}\right)\mathrm{d}y_{L_i}\mathrm{d}x_{L_i}
\end{aligned}
$$

where

$$
\mathbf{W}_{R_i}\left(x_{R_i},y_{R_i}\right)=\left[W_{R_i,1}\left(x_{R_i},y_{R_i}\right),\ldots,W_{R_i,n}\left(x_{R_i},y_{R_i}\right)\right]
$$

$$
\mathbf{W}_{L_i}\left(x_{L_i},y_{L_i}\right)=\left[W_{L_i,1}\left(x_{L_i},y_{L_i}\right),\ldots,W_{L_i,n}\left(x_{L_i},y_{L_i}\right)\right]
$$

$$\theta_0^{(x)} = \left[\theta_{0,1}^{(x)}, \ldots, \theta_{0,n}^{(x)}\right], \; \theta_0^{(y)} = \left[\theta_{0,1}^{(y)}, \ldots, \theta_{0,n}^{(y)}\right], \; \theta_0^{(z)} = \left[\theta_{0,1}^{(z)}, \ldots, \theta_{0,n}^{(z)}\right]$$

**Appendix B**

The stiffness and damping matrices $K$ and $C$ are as follows:

$$K = \begin{bmatrix} \mathbf{0}_{6\times 6} & \mathbf{0}_{6\times n} \\ \mathbf{0}_{n\times 6} & K_{77} \end{bmatrix}, \; C = \begin{bmatrix} \mathbf{0}_{6\times 6} & \mathbf{0}_{6\times n} \\ \mathbf{0}_{n\times 6} & C_{77} \end{bmatrix}$$

where

$$
\begin{aligned}
K_{77} &= \Re + \ell \\
\Re &= D\sum_{i=1}^{N}\int_0^a\int_{-b}^b \frac{\partial^2 \mathbf{W}_{R_i}^{\mathrm{T}}}{\partial x_{R_i}^2}\frac{\partial^2 \mathbf{W}_{R_i}}{\partial x_{R_i}^2}\mathrm{d}y_{R_i}\mathrm{d}x_{R_i} \\
&\quad + vD\sum_{i=1}^{N}\int_0^a\int_{-b}^b \frac{\partial^2 \mathbf{W}_{R_i}^{\mathrm{T}}}{\partial x_{R_i}^2}\frac{\partial^2 \mathbf{W}_{R_i}}{\partial y_{R_i}^2} + \frac{\partial^2 \mathbf{W}_{R_i}^{\mathrm{T}}}{\partial y_{R_i}^2}\frac{\partial^2 \mathbf{W}_{R_i}}{\partial x_{R_i}^2}\mathrm{d}y_{R_i}\mathrm{d}x_{R_i} \\
&\quad + 2(1-v)D\sum_{i=1}^{N}\int_0^a\int_{-b}^b \frac{\partial^2 \mathbf{W}_{R_i}^{\mathrm{T}}}{\partial x_{R_i}\partial y_{R_i}}\frac{\partial^2 \mathbf{W}_{R_i}}{\partial x_{R_i}\partial y_{R_i}}\mathrm{d}y_{R_i}\mathrm{d}x_{R_i} \\
&\quad + D\sum_{i=1}^{N}\int_0^a\int_{-b}^b \frac{\partial^2 \mathbf{W}_{R_i}^{\mathrm{T}}}{\partial y_{R_i}^2}\frac{\partial^2 \mathbf{W}_{R_i}}{\partial y_{R_i}^2}\mathrm{d}y_{R_i}\mathrm{d}x_{R_i} \\
&\quad + k\sum_{i=1}^{N}\left(\Delta\Theta_{R_{A_i}}\right)^{\mathrm{T}}\Delta\Theta_{R_{A_i}} + k\sum_{i=1}^{N}\left(\Delta\Theta_{R_{B_i}}\right)^{\mathrm{T}}\Delta\Theta_{R_{B_i}}
\end{aligned}
$$

where

$$\Delta\Theta_{R_{A_1}} = \left.\frac{\partial \mathbf{W}_{R_1}^{\mathrm{T}}}{\partial x_{R_1}}\right|_{\substack{x_{R\,1}=0\\ y_{R\,1}=y_a}} \left.\frac{\partial \mathbf{W}_{R_1}}{\partial x_{R_1}}\right|_{\substack{x_{R\,1}=0\\ y_{R\,1}=y_a}}, \; \Delta\Theta_{R_{B_1}} = \left.\frac{\partial \mathbf{W}_{R_1}^{\mathrm{T}}}{\partial x_{R_1}}\right|_{\substack{x_{R\,1}=0\\ y_{R\,1}=y_b}} \left.\frac{\partial \mathbf{W}_{R_1}}{\partial x_{R_1}}\right|_{\substack{x_{R\,1}=0\\ y_{R\,1}=y_b}}$$

$$\Delta\Theta_{L_{A_1}} = \left.\frac{\partial \mathbf{W}_{L_1}^{\mathrm{T}}}{\partial x_{L_1}}\right|_{\substack{x_{L\,1}=0\\ y_{L\,1}=y_a}} \left.\frac{\partial \mathbf{W}_{L_1}}{\partial x_{L_1}}\right|_{\substack{x_{L\,1}=0\\ y_{L\,1}=y_a}}, \; \Delta\Theta_{L_{B_1}} = \left.\frac{\partial \mathbf{W}_{L_1}^{\mathrm{T}}}{\partial x_{L_1}}\right|_{\substack{x_{L\,1}=0\\ y_{L\,1}=y_b}} \left.\frac{\partial \mathbf{W}_{L_1}}{\partial x_{L_1}}\right|_{\substack{x_{L\,1}=0\\ y_{L\,1}=y_b}}$$

when $i = 2, 3, \ldots, N$,

$$\Delta\Theta_{R_{A_i}} == \left.\frac{\partial \mathbf{W}_{R_i}}{\partial x_{R_i}}\right|_{\substack{x_{R_i}=0\\ y_{R_i}=y_a}} - \left.\frac{\partial \mathbf{W}_{R_{i-1}}}{\partial x_{R_{i-1}}}\right|_{\substack{x_{R_{i-1}}=a\\ y_{R_{i-1}}=y_a}}, \; \Delta\Theta_{R_{B_i}} == \left.\frac{\partial \mathbf{W}_{R_i}}{\partial x_{R_i}}\right|_{\substack{x_{R_i}=0\\ y_{R_i}=y_b}} - \left.\frac{\partial \mathbf{W}_{R_{i-1}}}{\partial x_{R_{i-1}}}\right|_{\substack{x_{R_{i-1}}=a\\ y_{R_{i-1}}=y_b}}$$

$$\Delta\Theta_{L_{A_i}} == \left.\frac{\partial \mathbf{W}_{L_i}}{\partial x_{L_i}}\right|_{\substack{x_{L_i}=0\\ y_{L_i}=y_a}} - \left.\frac{\partial \mathbf{W}_{L_{i-1}}}{\partial x_{L_{i-1}}}\right|_{\substack{x_{L_{i-1}}=a\\ y_{L_{i-1}}=y_a}}, \; \Delta\Theta_{L_{B_i}} == \left.\frac{\partial \mathbf{W}_{L_i}}{\partial x_{L_i}}\right|_{\substack{x_{L_i}=0\\ y_{L_i}=y_b}} - \left.\frac{\partial \mathbf{W}_{L_{i-1}}}{\partial x_{L_{i-1}}}\right|_{\substack{x_{L_{i-1}}=a\\ y_{L_{i-1}}=y_b}}$$

The expression of $\ell$ can be obtained from $\Re$ in the last equation by replacing $R_i$ with $L_i$. The damping matrix $C_{77}$ is as follows:

$$C_{77} = \kappa_M \mathbf{M} + \kappa_K \mathbf{K}_p + c\mathbf{C}_j$$

The coefficients $\kappa_M$ and $\kappa_K$ are the proportionality constants.

where

$$
\begin{aligned}
\boldsymbol{K}_p &= \Re_p + \ell_p \\
\Re_p &= D \sum_{i=1}^{N} \int_0^a \int_{-b}^b \frac{\partial^2 \mathbf{W}_{R_i}^{\mathrm{T}}}{\partial x_{R_i}^2} \frac{\partial^2 \mathbf{W}_{R_i}}{\partial x_{R_i}^2} \mathrm{d}y_{R_i}\mathrm{d}x_{R_i} \\
&\quad + vD \sum_{i=1}^{N} \int_0^a \int_{-b}^b \frac{\partial^2 \mathbf{W}_{R_i}^{\mathrm{T}}}{\partial x_{R_i}^2} \frac{\partial^2 \mathbf{W}_{R_i}}{\partial y_{R_i}^2} + \frac{\partial^2 \mathbf{W}_{R_i}^{\mathrm{T}}}{\partial y_{R_i}^2} \frac{\partial^2 \mathbf{W}_{R_i}}{\partial x_{R_i}^2} \mathrm{d}y_{R_i}\mathrm{d}x_{R_i} \\
&\quad + 2(1-v)D \sum_{i=1}^{N} \int_0^a \int_{-b}^b \frac{\partial^2 \mathbf{W}_{R_i}^{\mathrm{T}}}{\partial x_{R_i}\partial y_{R_i}} \frac{\partial^2 \mathbf{W}_{R_i}}{\partial x_{R_i}\partial y_{R_i}} \mathrm{d}y_{R_i}\mathrm{d}x_{R_i} \\
&\quad + D \sum_{i=1}^{N} \int_0^a \int_{-b}^b \frac{\partial^2 \mathbf{W}_{R_i}^{\mathrm{T}}}{\partial y_{R_i}^2} \frac{\partial^2 \mathbf{W}_{R_i}}{\partial y_{R_i}^2} \mathrm{d}y_{R_i}\mathrm{d}x_{R_i}
\end{aligned}
$$

The expression of $\ell_p$ can be obtained from $\Re_p$ by replacing $R_i$ with $L_i$.

$$
C_j = \sum_{i=1}^{N} \left( \Delta\Theta_{R_{A_i}} \right)^{\mathrm{T}} \Delta\Theta_{R_{A_i}} + \sum_{i=1}^{N} \left( \Delta\Theta_{R_{B_i}} \right)^{\mathrm{T}} \Delta\Theta_{R_{B_i}} \\
+ \sum_{i=1}^{N} \left( \Delta\Theta_{L_{A_i}} \right)^{\mathrm{T}} \Delta\Theta_{L_{A_i}} + \sum_{i=1}^{N} \left( \Delta\Theta_{L_{B_i}} \right)^{\mathrm{T}} \Delta\Theta_{L_{B_i}}
$$

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
