# Peer review of "Dynamic Modeling and Attitude–Vibration Cooperative Control for a Large-Scale Flexible Spacecraft"

_actuators, doi:10.3390/act12040167_

Round 1

Reviewer 1 Report

1 List of reference specially in the area of deriving the dynamic model of large spacecraft needs to be updated for instance

a.       Modal truncation for flexible spacecraft, PC Hughes, RE Skelton - Journal of Guidance and Control, 1981 - arc.aiaa.org

b.       Guy, Nicolas, et al. "Dynamic modeling and analysis of spacecraft with variable tilt of flexible appendages." Journal of Dynamic Systems, Measurement, and Control 136.2 (2014): 021020.

c.       Azimi, Milad, and Eshagh Farzaneh Joubaneh. "Dynamic modeling and vibration control of a coupled rigid-flexible high-order structural system: A comparative study." Aerospace Science and Technology 102 (2020): 105875.

2The authors did not indicate the number of the solar panel (N) in the simulation to claim that the spacecraft is large.

3 The authors need to clearly state the novelty presented in this paper, specifically in the deriving the dynamic model and the use of the cooperative control with respect to already published articles.

4 The authors claim that the derived dynamic model is accurate and simple. Even thou this is a paradox; the authors need at least to compare the open loop model to existing models already to prove the claim stated.

Reviewer 2 Report

This a very interesting paper about a new model for a large and flexible spacecraft and its control using three different strategies. These are my comments:

In abstract section, you could try to depurate the first two sentences. They are a little redundant. For example, in only these first two sentences, you use the word “spacecraft” 5 times.

You should correct all your typos throughout your paper.

Although, I can understand your English, it is very advisable the complete review of the paper by a native speaker. Also, the style can be improved.

In abstract section (lines 12 and 13), you wrote “The analytic global modes are used to discrete the rigid-flexible coupling dynamic model” Are you using the word “discrete” as a verb. Up to the best of my knowledge, this word is only an adjective.

In line 428, you wrote: “with different numbers of modes are worked out through Eq. (22) numerically, as shown”. However, Eq. (22) is associated with the control torque, ¿is this correct?

In line 476, you wrote: “Q is a semipositive-definite matrix”. The correct expression is “positive semidefinite matrix”

In Equation (21). How can you guarantee that this Riccati equation has a solution?

In line 548, you mentioned: “Let R = 1, Q = diag(5,1,1,5000,1,1) ” Why are you specifically using these values? How can you select them? Why do you use only a diagonal positive matrix Q instead of a positive semidefinite matrix Q?

In line 563, you wrote: “The PD gains of PD controller are taken as Kd =48 and Kp=1.2, respectively.” Why did you select specifically these values? You should detailly explain your criterion of selection. Now, a very important point, if you are only using trial and error, you should present a resume of all your trials.

I don’t find the parameters for PD+IS controller. Are the parameters for the PD component (of the PD+IS controller) the same as in line 563? If this is the case, you should explicitly mention it.

A very important question: are you testing your controllers against the discrete model? If this is the case, to see the actual performance of your controllers, you should test them against a complete distributed parameter model of the spacecraft. The discrete model can be used only for controller design purposes.

Reviewer 3 Report

The following comments can be useful to improve the quality of the manuscript paper entitled “Dynamic modeling and attitude-vibration cooperative control for a large-scale flexible spacecraft”:

1) A simplified and accurate dynamic model of the system for purposes of control is proposed. It is suggested to explain why the nonlinear dynamics can be neglected in the reduced mathematical model (14).

2) Include more information about how the real-time control inputs can be generated for real-life implementation.

3) As described in the paper, external disturbances have been considered in other control design methodologies. In this work, LQR, PD and PD+IS algorithm are preferred for control design. More detailed information about the advantages with respect to other important control techniques published in journals to suppress time-varying disturbances can be added.

4) In this paper, the simplified mathematical model (14) has been used for control design. There are several kinds of disturbances, for instance, resonant forced excitations, unmodeled dynamics, uncertainties, etc. It is suggested to include in the state space representation the time-varying disturbances.

5) It is suggested to describe the main advantages and differences of the presented techniques for controller design with respect to the other ways for suppressing disturbances: a robust active control scheme for automotive engine vibration based on disturbance observer; output feedback dynamic control for trajectory tracking and vibration suppression; disturbance feedforward control for active vibration isolation systems with internal isolator dynamics.

6) In Line 563, the PD gains of PD controller are taken as Kd =48 and Kp=1.2, respectively, Provide more information about how the control gains were selected. Include more information about the control robustness analysis for different variable disturbances.

7) Please include some comments about how the presented control design strategy based on mathematical model cold be extended to other practical dynamic systems.

Round 2

Reviewer 1 Report

The revision version is good to be published. 

Author Response

Thanks for your approval of our research. Your comments are all valuable and very helpful for revising and improving our paper. Also, your comments make us thinking seriously about things we have never thought about before. Following your comments, we have revised our manuscript. We wish that this revised manuscript will meet the high standard requirements for publication.

Reviewer 2 Report

The authors of this paper have followed all my comments. I recommend the acceptance of this paper.

Author Response

(The authors gave the same response as above.)

Reviewer 3 Report

Quality of the figures should be improved.

Author Response

Thanks for your approval of our research. Your comments are all valuable and very helpful for revising and improving our paper. Also, your comments make us thinking seriously about things we have never thought about before. Following your comments, we have revised our manuscript. The quality of the figures in the manuscript have been improved. According to your comments, we have replotted the Fig.15-Fig.17. We wish that this revised manuscript will meet the high standard requirements for publication.